# Propionic acid toxicity and utilization of α-ketobutyric acid in *Neisseria meningitidis* via the methylcitrate cycle under specific conditions

Adelfia Talà,[1] Matteo Calcagnile,[2] Silvia Caterina Resta,[2] Salvatore Maurizio Tredici,[2] Giuseppe Egidio De Benedetto,[3] Cecilia Bucci,[2] Pietro Alifano[2]

ABSTRACT  *Neisseria meningitidis* is a human-specific, transient colonizer of the nasopharynx that occasionally causes invasive disease. It can utilize a limited range of compounds as primary carbon sources, including glucose, maltose, lactate, and pyruvate, which are present in varying concentrations in microenvironments relevant to meningococcal infection. Additionally, intermediates from the tricarboxylic acid cycle, such as succinate, fumarate, and malate, as well as amino acids like glutamate, are utilized as supplementary carbon sources. Notably, *N. meningitidis* also possesses a functional methylcitrate cycle (MCC), which enables the assimilation of propionic acid and mitigates its toxicity. In this study, we investigated propionate toxicity and MCC functionality in wild-type *N. meningitidis* strains and *prpB-*, *prpC-*, *ackA1-*, and *ackA2*-defective mutants under various growth conditions. We observed that propionate toxicity was influenced by the primary carbon source and additional factors, such as bicarbonate. Specifically, *prpB-* and *prpC*-defective mutants showed high sensitivity to propionate when cultured with glucose or pyruvate, but were not inhibited even by high concentrations of propionate when grown with lactate. The mechanisms underlying the conditional toxicity of propionate were further explored and discussed. Additionally, in the genome of 41 out of 128 *N. meningitidis* strains, we identified a gene encoding a transporter from the 4-toluene sulfonate uptake permease family, located between *prpC* and *acnD* in the MCC gene cluster. Genetic inactivation of this gene, named *kbuT*, impaired the ability to take up and oxidize α-ketobutyrate, an α-keto acid abundant in host cells, which can be used as a carbon source through the MCC.

IMPORTANCE  Meningococci are metabolically versatile organisms, switching between intracellular and extracellular lifestyle during colonization and invasive disease. Niche switching impacts on how bacteria communicate with host to find a balance between nutrient assimilation and protection against toxicity of some metabolites. The methylcitrate pathway fulfills this function, providing a compromise between propionate assimilation and propionate detoxification, in relation to the colonized host microenvironments. In this study, we revealed an unexpected difference in the sensitivity of meningococci to propionate when grown with different carbon sources. We also characterized the function of a gene located within the prp operon that encodes a transporter of α-ketobutyrate, an α-ketoacid abundant in host cells. These results contribute to extending our understanding of the metabolic adaptation mechanisms, which are crucial for meningococcal infection and virulence within the host microenvironments.

KEYWORDS  *Neisseria meningitidis*, methylcitrate cycle, propionic acid toxicity, α-ketobutyric acid metabolism

Editor Ana-Maria Dragoi, LSU Health Shreveport, Shreveport, Louisiana, USA

Address correspondence to Pietro Alifano, pietro.alifano@unisalento.it.

The authors declare no conflict of interest.

See the funding table on p. 23.

The ability of a particular microorganism to occupy a specific ecological niche within the host, whether or not causing invasive disease, is regulated by a multitude of factors, including its metabolic capabilities (1, 2). A crucial factor in the commensal and pathogenic behavior of *Neisseria* species is their capacity to obtain nutrients essential for their survival in the different microenvironments of the human host (3, 4).

*Neisseria meningitidis* can use only a small number of carbon sources, including the two carbohydrates maltose and glucose (5), lactate (6, 7), and a few amino acids, such as glutamate (8). In addition, pyruvate and the tricarboxylic acid (TCA) cycle intermediates, succinate, fumarate, and malate, are also assimilated and catabolized by meningococci (9, 10).

Lactate is transported into meningococci by lactate permease LctP (11), and lactate metabolism is accomplished by at least three lactate dehydrogenases: two membrane-bound flavin mononucleotide-containing lactate dehydrogenases, LldD and LdhD, which catalyze the oxidation of *L*-lactate and *D*-lactate, respectively, and a soluble $NAD^+$-dependent *D*-lactate dehydrogenase, LdhA (12–15). Lactate provides energy for growth by providing electrons to the electron transport chain when it is oxidized to pyruvate. Pyruvate then generates acetyl-CoA, the precursor of fatty acid synthesis, and constituents and products of the TCA cycle that are entirely functional under this growth condition (9, 15, 16).

In sharp contrast with many bacteria that use the phosphotransferase system to transport efficiently the glucose into the bacteria (17), the meningococcus transports glucose via ion symport permease GluP (also named GlcP) (18). In meningococci, glucose is metabolized largely via the Entner-Doudoroff pathway, which generates relatively small amounts of energy, with a little contribution from the pentose phosphate pathway (9). At neutral pH values, the catabolism of glucose results in the accumulation of acetate, which is not further catabolized until glucose is depleted or growth becomes limiting. Indeed, growth on glucose markedly reduces the levels of TCA cycle enzymes in gonococci and meningococci (19–21). To sustain aerobic metabolism, the activity of the TCA cycle may be supported by the TCA cycle intermediates succinate, fumarate, malate, and α-ketoglutarate (9, 10), and by glutamate which can be directly converted to α-ketoglutarate (8, 22–24).

These carbon source biochemical pathways and the regulatory mechanisms concerned have wider implications because glucose, lactate, pyruvate, and glutamate are present at very different ratios in microenvironments relevant to meningococcal infection (15). Glucose is the predominant carbon source in blood; by contrast, lactate is the major carbon source during growth in the cerebrospinal fluid (11), as well as in the saliva and in mucosal environments that are colonized by lactic bacteria, such as the oropharynx and the nasopharynx (25, 26). Lactate and pyruvate tend to be used as major carbon (energy) sources within phagocytic cells (15, 27). Glutamate may represent an important carbon (and nitrogen) source both in the intracellular environment and also in cerebrospinal fluid where the levels of this amino acid strongly increase during meningitis (28, 29) and correlate with disease severity (30). The uptake of L-glutamate is considered critical for meningococcal infection in both cell (23, 31) and animal infection (4, 22, 31) models and is also instrumental in preventing oxidative injury, as L-glutamate is the precursor of glutathione (32). Finally, there is no evidence that meningococcus can utilize glucose-6-phosphate, which is an available carbon source in the intracellular environment, as other pathogenic bacteria do (33).

A genomic island allows *N. meningitidis* to utilize propionic acid by the methylcitrate cycle (MCC) under conditions of poor nutrient growth and to overcome toxicity of propionic acid (34). This genomic island, containing the genes *prpB*, *prpC*, *acnD*, *prpF*, and *ackA2*, is absent in *N. lactamica*, and this would confer a selective advantage to *N. meningitidis* over *N. lactamica* in young adults (34). Indeed, *N. lactamica* and *N. meningitidis* have age-related colonization patterns (35–37), and it has been observed that in the pharynx of young adults, the increase in meningococcal carriage rate is correlated with an increase in the abundance of bacteria producing propionic acid. This organic acid

is toxic to many microorganisms (38) and can be used instead as an additional carbon source by *N. meningitidis*, but not by *N. lactamica* (34).

These premises led us to investigate some aspects related to the toxicity of propionic acid in meningococcus, in relation to the main carbon source used for growth, and to explore new possible functions of the MCC in this microorganism, such as the use of α-ketobutyrate, an α-keto acid at the intersection of many metabolic pathways in the meningococcus and the infected host cell. It is involved in the metabolism of many amino acids (glycine, cysteine, methionine, valine, leucine, serine, threonine, isoleucine) and also plays a role in propionate metabolism and C-5 branched dibasic acid metabolism.

## RESULTS

### The growth inhibition by propionate depends on the main carbon source for meningococcal growth

To investigate the role of MCC under different growth conditions, *prpB* (encoding the 2-methylisocitrate lyase; NMB430 in strain MC58) or *prpC* (encoding the 2-methylcitrate synthase; NMB431 in strain MC58) was insertionally inactivated in *N. meningitidis* B1940 by pDE△*prpB* or pDE△*prpC*, respectively (Fig. S1A and B), and the resulting *prpB*- and *prpC*-defective mutants were genetically complemented (Fig. S2A and B). The growth curves of bacteria grown in different media were evaluated by determining the optical density at 600 nm ($OD_{600\ nm}$).

In the absence of propionate, the *prpB*- and *prpC*-defective mutants behaved like the wild-type strain when growing in meningococcal-defined agar (MCDA) with glucose, pyruvate, or lactate as the main carbon source (Fig. 1; File S1). In contrast, growth of the mutants and not that of the wild-type strain and *prpB*- and *prpC*-complemented strains was inhibited when 1 mM propionate was added to MCDA-glucose or MCDA-pyruvate (Fig. 1; Fig. S2C; File S2), whereas all strains grew in MCDA-lactate with 1 mM propionate (Fig. 1). Furthermore, consistent with previous findings (34), we found that the final biomass of the wild-type strain was increased in MCDA-pyruvate supplemented with subinhibitory concentrations of propionate. At 5 mM propionate, all three strains did not grow in MCDA-glucose or MCDA-pyruvate, while they grew in MCDA-lactate, and, unexpectedly, in this medium, the growth of the wild-type strain appeared slightly inhibited compared to that of the *prpB*- and *prpC*-defective mutants and even more so in the presence of 10 mM propionate. This result suggested that the mechanism of propionate inhibition in bacteria growing on glucose was different from that in bacteria growing on lactate as the main carbon source.

Evaluation of colony-forming units (CFU) of *N. meningitidis* B1940 grown in MCDA-glucose or MCDA-pyruvate in the absence or presence of 5 mM propionate demonstrated that propionate did not kill the bacteria but only prevented their growth, because the CFU number of the initial inoculum remained almost stable for up to 12 h in the media with the addition of propionate (Fig. S3).

As reported previously, in MCDA-glucose or MCDA-pyruvate, the growth of the wild-type strain B1940 and *prpB*- and *prpC*-defective mutants was completely inhibited in the presence of 5 mM propionate. However, it was previously shown that both the wild-type strain MC58 and a *prpC*-defective isogenic mutant grew with glucose or pyruvate in the presence of 5 mM propionic acid in a different chemically defined medium (referred to as CDM) (34). We hypothesized that this discrepancy could be due to differences in the chemical composition of MCDA and CDM (Table S1). The main differences between CDM and MCDA are as follows: (i) the presence of L-glutamine (4 mM) in CDM instead of L-glutamic acid (8 mM) in MCDA; (ii) a higher L-serine concentration (4.75 mM) in CDM than in MCDA (0.2 mM); the presence of a high L-cystine concentration in CDM (3.8 mM) instead of a low L-cysteine concentration in MCDA (0.06 mM); the presence of $NaHCO_3$ (10 mM) only in the CDM. To understand which of these differences might affect growth in the presence of propionate, the MCDA medium was modified by adding, one by one, 3.8 mM L-cysteine, 4 mM L-glutamine,

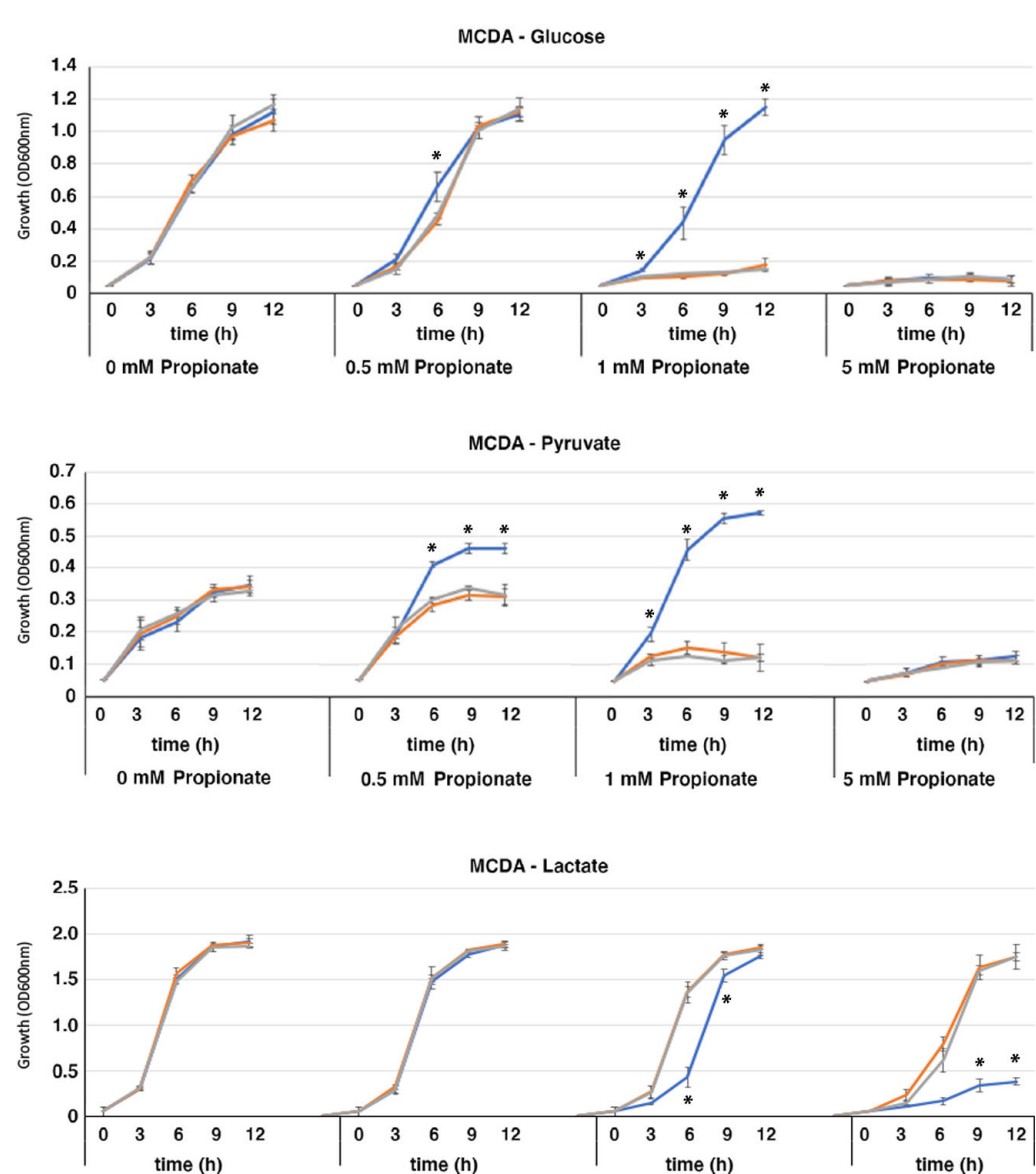

**FIG 1** Growth curves of *N. meningitidis* B1940 and isogenic *prpB*- and *prpC*-defective mutants in MCDA medium with different carbon sources. Growth of *N. meningitidis* B1940 and *prpB*- and *prpC*-defective mutants in MCDA-glucose (top), MCDA-pyruvate (middle), or MCDA-lactate (bottom), in the absence or in the presence of different concentrations of propionate. Three independent *prpB*-defective mutants and one *prpC*-defective mutant were tested, along with the wild-type B1940, in three independent experiments as reported in File S1. Means and standard deviations are shown at each time point. Asterisks indicate statistically significant differences ($P < 0.05$) between B1940 and *prpB*- or *prpC*-defective mutants at the corresponding time points.

4.75 mM L-serine, or 10 mM $NaHCO_3$. The growth of the wild-type strain B1940 was then analyzed in the modified media, in the presence of 5 mM propionate, using glucose as the main carbon source (Fig. 2A; File S1). We found that the strain was able to grow in MCDA-glucose with 5 mM propionate only when the medium was supplemented with 10 mM $NaHCO_3$. In this modified medium, the defective mutants *prpB* and *prpC*

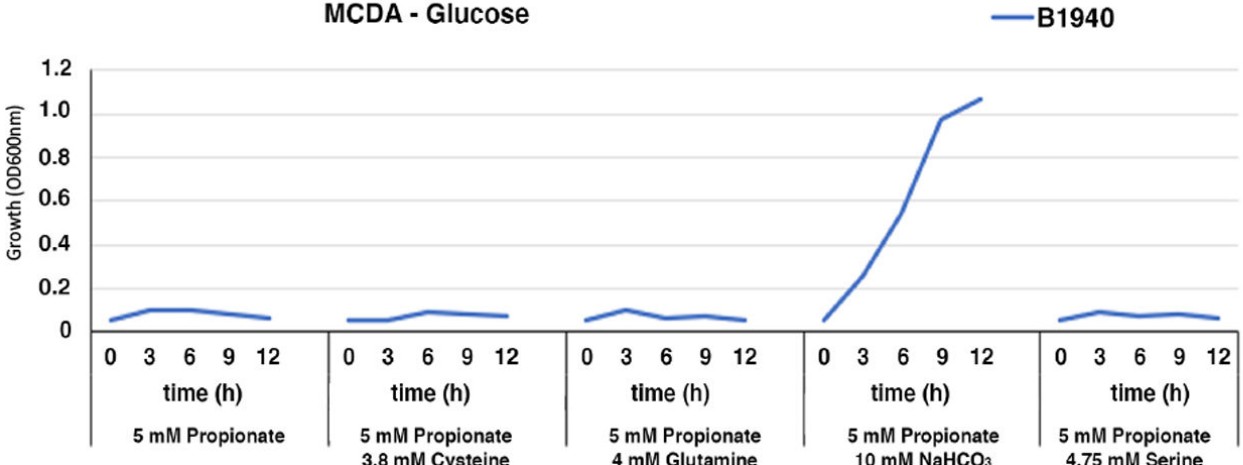

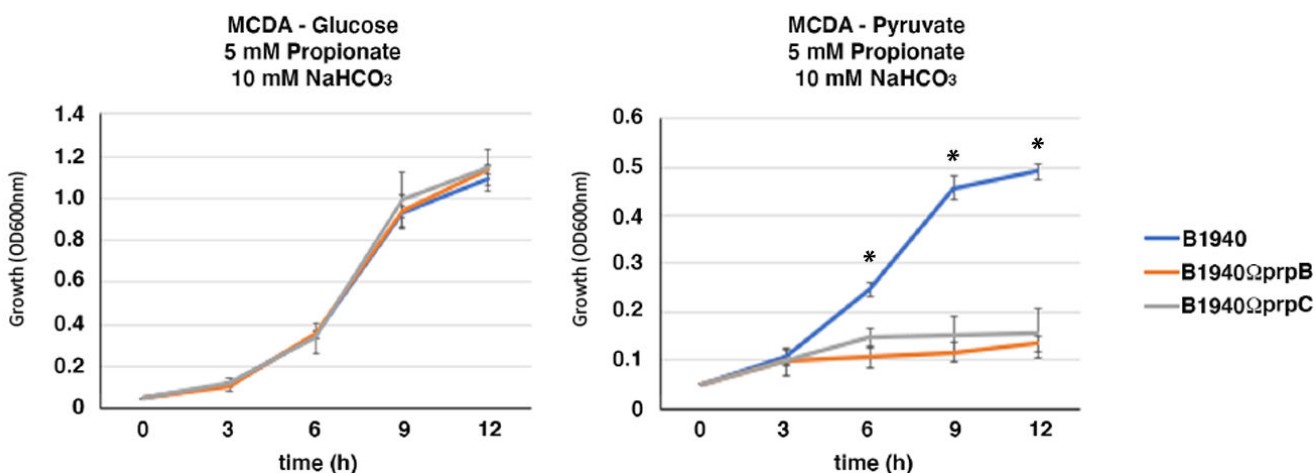

FIG 2 Effects of different supplements on the growth of *N. meningitidis* B1940 and isogenic *prpB*- and *prpC*-defective mutants in MCDA-glucose with or without propionate. (A) Effects of different supplements (3.8 mM L-cysteine, 4 mM L-glutamine, 10 mM NaHCO$_3$, and 4.75 mM L-serine) on the growth of *N. meningitidis* B1940 in MCDA-glucose in the presence of 5 mM propionate. (B) Growth curves of *N. meningitidis* B1940 and *prpB*- and *prpC* derivative mutants in MCDA-glucose and MCDA-pyruvate supplemented with 10 mM NaHCO$_3$ and 5 mM propionate. Three independent *prpB*-defective mutants and one *prpC*-defective mutant were tested, along with the wild-type B1940, in three independent experiments as reported in File S1. Means and standard deviations are shown at each time point. Asterisks indicate statistically significant differences ($P < 0.05$) between B1940 and *prpB*- or *prpC*-defective mutants, at the corresponding time points.

also grew, while in MCDA-pyruvate containing 5 mM propionate and 10 mM NaHCO$_3$, only the wild-type strain, but not the mutants, grew (Fig. 2A; File S1). These results demonstrated that NaHCO$_3$ was able to attenuate the growth inhibition by propionate.

## Distinct mechanisms cause growth inhibition by propionate on different main carbon sources

Propionate toxicity has been attributed to many different mechanisms (38) including non-specific mechanisms due to propionate itself leading to decreased intracellular pH,

anion accumulation, and dissipation of proton motive force (39), and specific mechanisms due to the accumulation of propionyl-CoA (38). Propionyl-CoA is a well-established inhibitor of the pyruvate dehydrogenase in bacterial and fungal systems (40, 41); it is a competitive inhibitor of citrate synthase (42), and in *Salmonella enterica,* it can be a substrate of citrate synthase leading to production of a toxic isomer of 2-methylcitrate (not produced by the methylcitrate synthase) which inhibits the fructose-1,6-bisphosphatase, a key enzyme in gluconeogenesis (43, 44). Other studies also suggest that the inhibitory effects of high propionate concentration may be caused in part by 2-methylisocitrate accumulation inhibiting the NADP-dependent isocitrate dehydrogenase, such as in *Aspergillus nidulans* (45, 46), depletion of oxaloacetate from the TCA cycle and gluconeogenic pathways (47), and altered thiol metabolism due to homocysteine accumulation (48). In fact, thiol (glutathione, cysteine, and methionine), aspartate, glycine, serine, and threonine metabolisms are connected with the MCC through the O-succinyl-L-homoserine metabolic node. More recently, propionic acid has been shown to inactivate alanine racemase, leading to activation of the Rcs stress response system in *Serratia marcescens* (49).

To dissect between these different mechanisms, we generated an *ackA2*-defective mutant (Fig. S4), impaired in the synthesis of propionyl-CoA (34), and analyzed the growth of the mutant in MCDA-glucose, MCDA-pyruvate, or MCDA-lactate in the absence or in the presence of different concentrations of propionate (Fig. 3A). *ackA2* (also indicated as *ackA-1*; NMB435 in strain MC58) is located in the *prp* locus (Fig. S1). The results demonstrated that the *ackA2* mutant was able to grow in MCDA-pyruvate containing 5 mM propionate and in MCDA-lactate containing 10 mM propionate, while the wild-type strain was unable to grow under these conditions (Fig. 3A; File S3), suggesting that the growth inhibition by propionate was due to the accumulation of propionyl-CoA or its metabolism.

Surprisingly, we observed no difference between the *ackA2*-defective mutant and the wild-type strain in MCDA-glucose with 5 mM propionate, as the growth of both strains was inhibited. This discrepancy could be attributed to the fact that *N. meningitidis* has two genes encoding *bona fide* acetate/propionate kinases, *ackA1* and *ackA2*, and *ackA2* is significantly downregulated when meningococci are cultured with glucose as the main carbon source (19). We hypothesized that under these conditions, propionyl-CoA biosynthesis may be primarily carried out by the acetate kinase/propionate kinase encoded by the *ackA1* (also known as *ackA-2*, NMB1518 in strain MC58). This hypothesis was supported by results of reverse transcriptase—real-time PCR, showing that *ackA2* mRNA levels were significantly higher in meningococci growing to the early log phase in MCDA-lactate or in MCDA-pyruvate compared to MCDA-glucose (about 11.9- and 4.3-fold higher, respectively) (Fig. 3B, left). In contrast, the *ackA1* mRNA levels were significantly lower in bacteria growing to the late log phase in MCDA-lactate or in MCDA-pyruvate compared to MCDA-glucose (about 6- and 5.8-fold lower, respectively) (Fig. 3B, left). The results of the reverse transcriptase—real-time PCR experiments also showed that *prpB* and *prpC* mRNA levels were significantly higher in bacteria growing to the early or late log phase in MCDA-lactate or in MCDA-pyruvate compared to MCDA-glucose (Fig. 3B, right), confirming that the MC gene cluster is significantly downregulated when meningococci are cultured with glucose as the main carbon source (19). Therefore, we generated an *ackA1*-defective mutant (Fig. S5A and B), and the results showed that this mutant grew in MCDA-glucose containing 5 mM propionate, confirming our hypothesis (Fig. 3A; File S3). We also noted that the *ackA1*-defective mutant grew in MCDA-pyruvate and MCDA-lactate in the presence of 5 mM and 10 mM propionate, respectively. This finding suggests that in MCDA-pyruvate and MCDA-lactate, both *ackA1* and *ackA2* contributed to growth inhibition by propionate, while in MCDA-glucose, only *ackA1* was involved in growth inhibition. We also found that in MCDA-pyruvate without propionate, both the *ackA1*- and the *ackA2*-defective mutants grew faster than the wild-type strain.

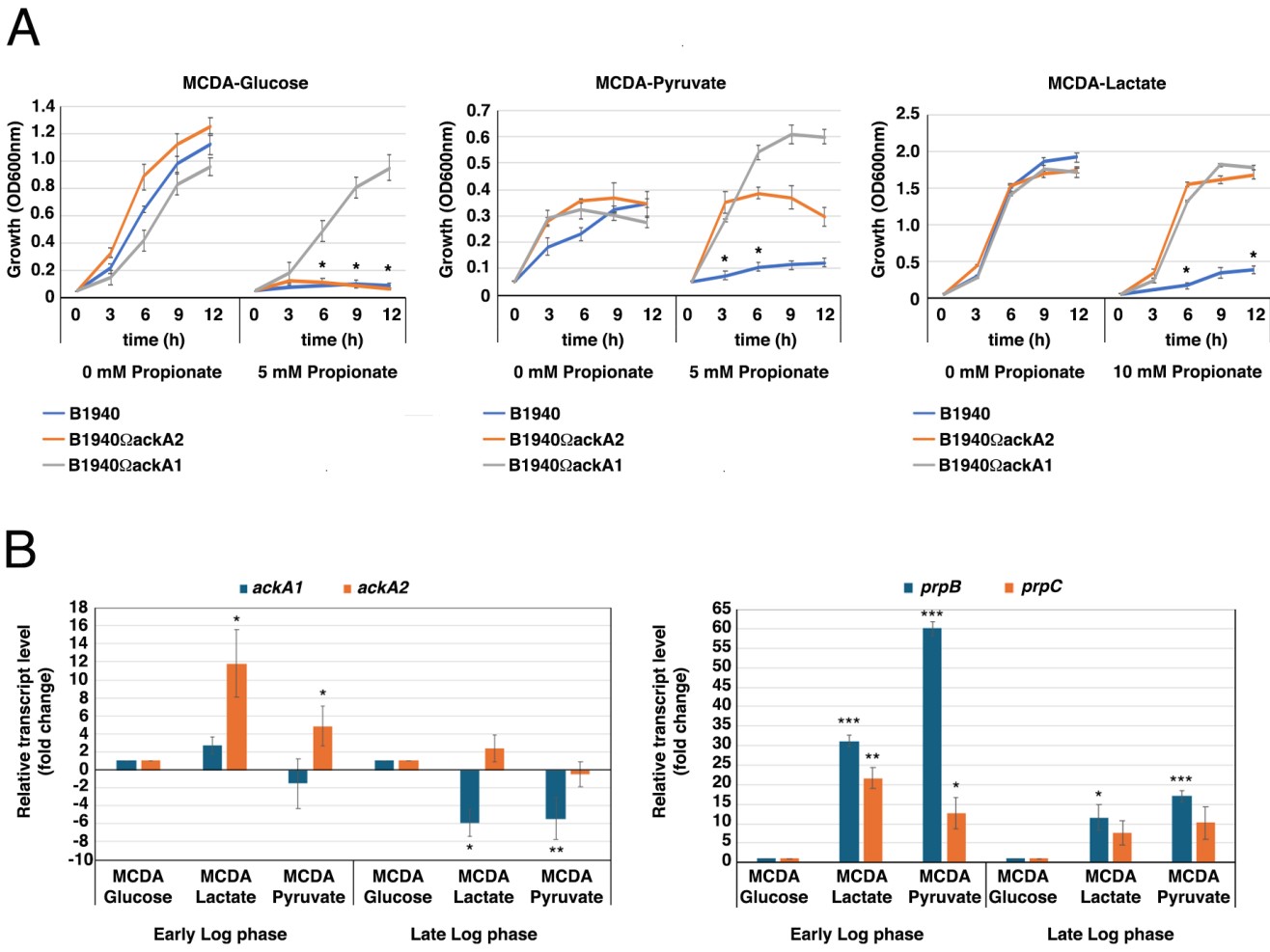

**FIG 3** Growth curves of *N. meningitidis* B1940 and isogenic *ackA1-* and *ackA2*-defective mutants in MCDA medium with different carbon sources, and expression of *prpB*, *prpC*, *ackA1*, and *ackA2*. (A) Growth of *N. meningitidis* B1940 and *ackA1-* and *ackA2*-defective mutants in MCDA-glucose (left), MCDA-pyruvate (middle), or MCDA-lactate (right), in the absence or in the presence of different concentrations of propionate. Three independent *ackA1-* and two independent *ackA2*-defective mutants were tested, along with the wild-type B1940, in independent experiments as reported in File S3. Means and standard deviations are shown at each time point. In MCDA-glucose, asterisks indicate statistically significant differences ($P < 0.05$) between B1940 and the *ackA1*-defective mutant at the corresponding time points. In MCDA-pyruvate and in MCDA-lactate, asterisks indicate statistically significant differences ($P < 0.05$) between B1940 and the *ackA1*-defective mutant and between B1940 and the *ackA2*-defective mutant, at the corresponding time points. (B) mRNA levels of *prpB*, *prpC*, *ackA1*, and *ackA2* were determined in *N. meningitidis* B1940 growing to early or late logarithmic growth phase in MCDA-glucose, MCDA-lactate, or MCDA-pyruvate. Asterisks indicate statistically significant differences (*$P < .05$; **$P < .01$; ***$P < .005$) with respect to the values determined in MCDA-glucose at the corresponding growth phases.

Regarding the other mechanisms responsible for growth inhibition by propionate, if the accumulation of 2-methylcitrate or 2-methylisocitrate were the main cause of propionate inhibition in MCDA-glucose, one would expect the *prpB* mutant to be more inhibited than the *prpC* mutant. In fact, the *prpB* mutant should accumulate 2-methylcitrate and 2-methylisocitrate (Fig. 4), as has been observed in *Pseudomonas aeruginosa* (50), while the *prpC* mutant does not. However, both mutants showed the same inhibition when grown on glucose (Fig. 1), making these hypotheses unlikely in *N. meningitidis*. Conversely, if the depletion of oxaloacetate from the TCA cycle and gluconeogenic pathways were primarily involved in growth inhibition by propionate, one would expect the wild-type strain to be more inhibited than the *prpB* and *prpC* mutants (Fig. 4), but this was not the case, at least when the bacteria were grown in MCDA-glucose or MCDA-pyruvate (Fig. 1). An altered thiol metabolism could also be excluded as the main mechanism, since intracellular glutathione levels determined in the

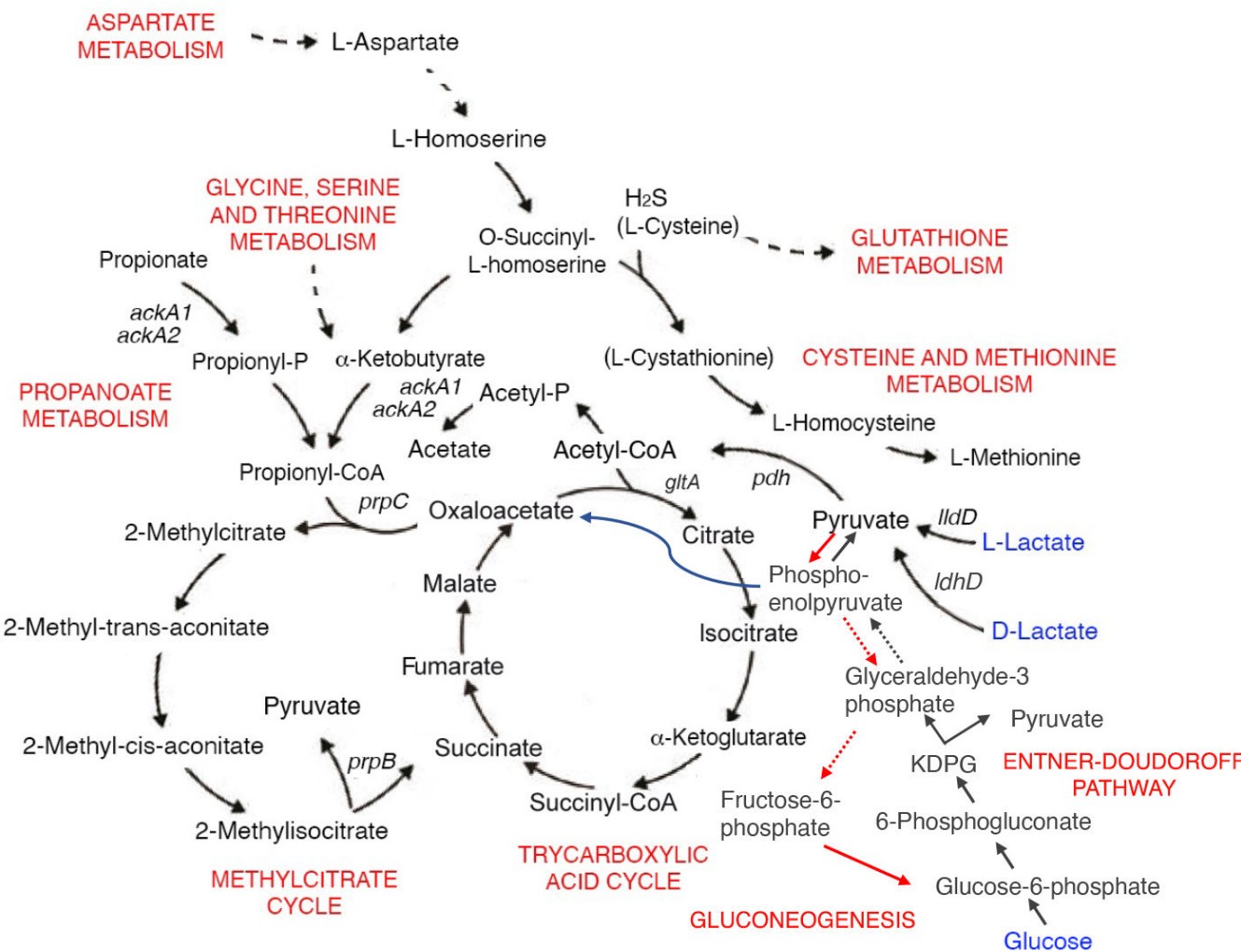

**FIG 4** Map of the central metabolism of *N. meningitidis* reconstructed based on KEGG metabolic pathways annotations.

wild-type and *prpB* and *prpC* mutants growing on MCDA-glucose or MCDA-lactate after the addition of 5 mM propionate and incubation for 1 h or without propionate addition demonstrated no significant differences ($P < 0.05$) between the strains (Fig. S6A and B).

Our data, instead, suggest that growth inhibition by propionate in *N. meningitidis* may be mediated by an accumulation of propionyl-CoA leading to an inhibition of pyruvate dehydrogenase and/or generation of a toxic isomer of 2-methylcitrate by citrate synthase, as shown in other microorganisms (40–44). Regarding the possible inhibition of pyruvate dehydrogenase, we found different intracellular levels of pyruvate in wild-type bacteria grown to the mid-log phase in MCDA-lactate (19.6 µM ± 2.2), MCDA-glucose (16.1 µM ± 0.9), and MCDA-pyruvate (3.9 µM ± 0.2) (Fig. S6C), which may contribute to explain the different sensitivity of the bacteria to propionate in the different media, as discussed below.

## The uptake of propionate is not mediated by the lactate permease

Intriguingly, we found that at 5 mM propionate, all three strains (the wild-type strain and the *prpB* and *prpC* mutants) were unable to grow in MCDA-glucose, while they grew in MCDA-lactate, and the question was as follows: why were the strains not sensitive to propionate when grown on lactate? One possible explanation was that the high lactate concentration could interfere with the entry of propionate. In fact, in the meningococcus, lactate enters through L-lactate permease LctP (NMB0543 in strain MC58) (11, 51), which

belongs to the LldP—lactate/glycolate-proton symporter superfamily, and it has been shown that some lactate-proton symporters are indifferent to the isomeric form of the substrate and are often capable of transporting also other short-chain monocarboxylates, such as pyruvate and propionate (52–54). Furthermore, several lactate transporters also function as antiporters, whereby the import of a substrate is linked to the export of a product (55), as in some lactic acid bacteria, which during malolactic fermentation import malate in exchange for lactate (56).

Thus, we generated an *lctP*-defective mutant (Fig. S7) and analyzed the growth of the mutant in MCDA-glucose or MCDA-pyruvate in the absence or in the presence of different concentrations of propionate. As expected, the mutant was unable to grow in MCDA-lactate, indicating that LctP is necessary to transport both L-lactate and D-lactate into the bacteria (Fig. 5; File S4). The results also demonstrated no difference in the growth between the *lctP* mutant and the wild-type strain in the presence of different concentrations of propionate, leading to the conclusion that lactate uptake does not interfere with propionate entry into the bacteria. Therefore, a mechanism of competition with lactate in the entry of propionate into the bacteria or a mechanism of propionate extrusion from the bacteria through the lactate permease was unlikely. This conclusion was supported by evidence that the extracellular propionate concentrations decreased during the growth of the wild-type strain in all media analyzed, including MCDA-lactate, indicating that propionate was consumed during growth with glucose, pyruvate, or lactate (Fig. 6).

## A gene encoding a putative α-ketobutyric acid transporter is located in the MCC gene cluster of several meningococcal strains

The analysis of 128 completely annotated genomes of *N. meningitidis* revealed that the region between *prpC* and *acnD* shows an intraspecies variation (Fig. 7). In 41 of the 128 completely annotated genomes analyzed, a gene encoding a 4-toluene sulfonate uptake permease (TSUP)-family protein of unknown function (NMB0432 in strain MC58) was mapped to this region (Fig. 7A). In contrast, in 11 genomes, two genes are mapped in this region: the first encodes a putative fatty acid desaturase belonging to the GntB_guanitoxin superfamily; the second encodes an unknown transporter of the drug/metabolite (DMT) family. In 75 genomes, a gene encoding a metallopeptidase of the M15 family was found in this region, followed by a gene encoding a DUF819-containing protein. Furthermore, in one strain, M22425, we found a genome rearrangement in the MCC gene cluster, probably promoted by the IS5-like element transposed downstream of *prpC* (Fig. 7B).

We focused on the gene encoding the TSUP family protein. TSUP (also called TauE/SafE/YfcA/DUF81 family) is a large family of proteins with only a few characterized members that are thought to catalyze the transport of various organic sulfur-containing compounds (57). Although genome context analyses initially supported the proposed role of sulfur-based compound transport for many TSUP homologs, the sequence diversity inherent in the TSUP family and further genome analyses suggested that members of this family may be involved in the transport of a broad range of compounds (57), including several amino acids and organic acids, such as glycine and acetic acid, as shown in *Shewanella oneidensis* (58). Furthermore, a gene highly homologous to that of *N. meningitidis* encoding the TSUP family protein (belonging to the same cluster, named Pas1) was found associated with the MCC gene cluster in *Photorhabdus asymbiotica* (57). This suggests the existence of a functional link between TSUP family proteins and the MCC.

Therefore, we generated a mutant in which the gene encoding the TSUP family protein (here referred to as *kbuT*) was insertionally inactivated, starting from the wild-type strain MC58 (Fig. S8A and B), and analyzed the growth of the mutant and the wild-type strains in MCDA-glucose or MCDA-lactate in the absence or in the presence of different concentrations of propionate (Fig. S8C; File S5). However, no differences were observed between the wild-type strain and the *kbuT*-defective mutant. Therefore,

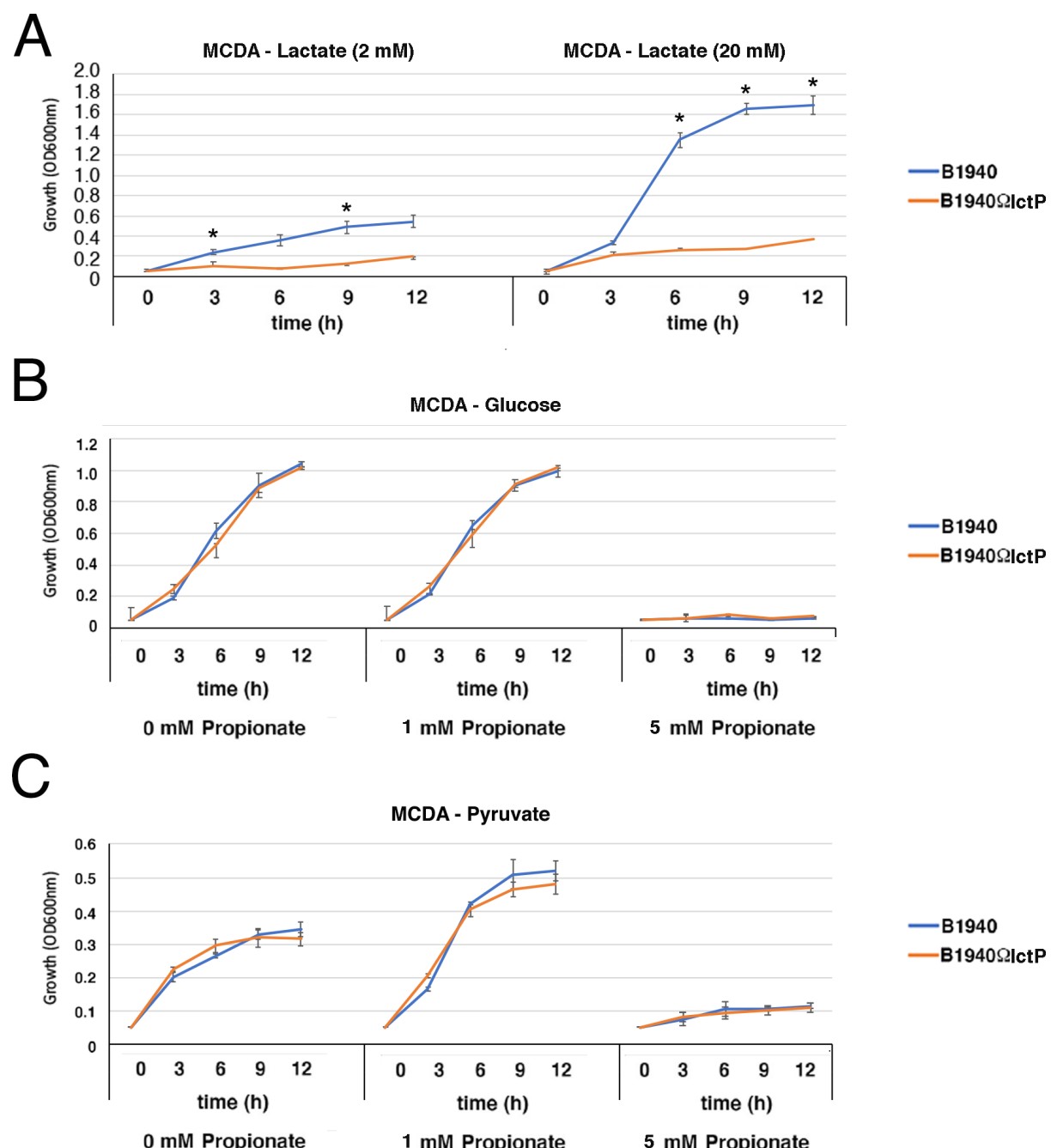

**FIG 5** Growth curves of *N. meningitidis* B1940 and its isogenic *lctP*-defective mutant in MCDA medium with different carbon sources. (A) Growth of *N. meningitidis* B1940 and *lctP*-defective mutant in MCDA-lactate, (B) MCDA-glucose, (C) MCDA-pyruvate, in the absence or in the presence of different concentrations of propionate. Two independent *lctP*-defective mutants were tested, along with the wild-type B1, in two independent experiments as reported in File S4. Means and standard deviations are shown at each time point. Asterisks indicate statistically significant differences ($P < 0.05$) between B1940 and the *lctP*-defective mutant at the corresponding time points.

considering that the gene inactivated in the *kbuT* mutant encodes a putative transporter, the BIOLOG Phenotype MicroArray system was used to search for possible defects in the oxidation of particular substrates in the mutant strain (Fig. 8A; File S6). In line with literature findings, the wild-type strain MC58 of *N. meningitidis* tested positive for the following substrates: α-D-glucose, D-maltose, L-aspartic acid, L-glutamic acid, L-lactic acid, α-ketoglutaric acid, L-malic acid, α-hydroxybutyric acid, α-ketobutyric acid, and

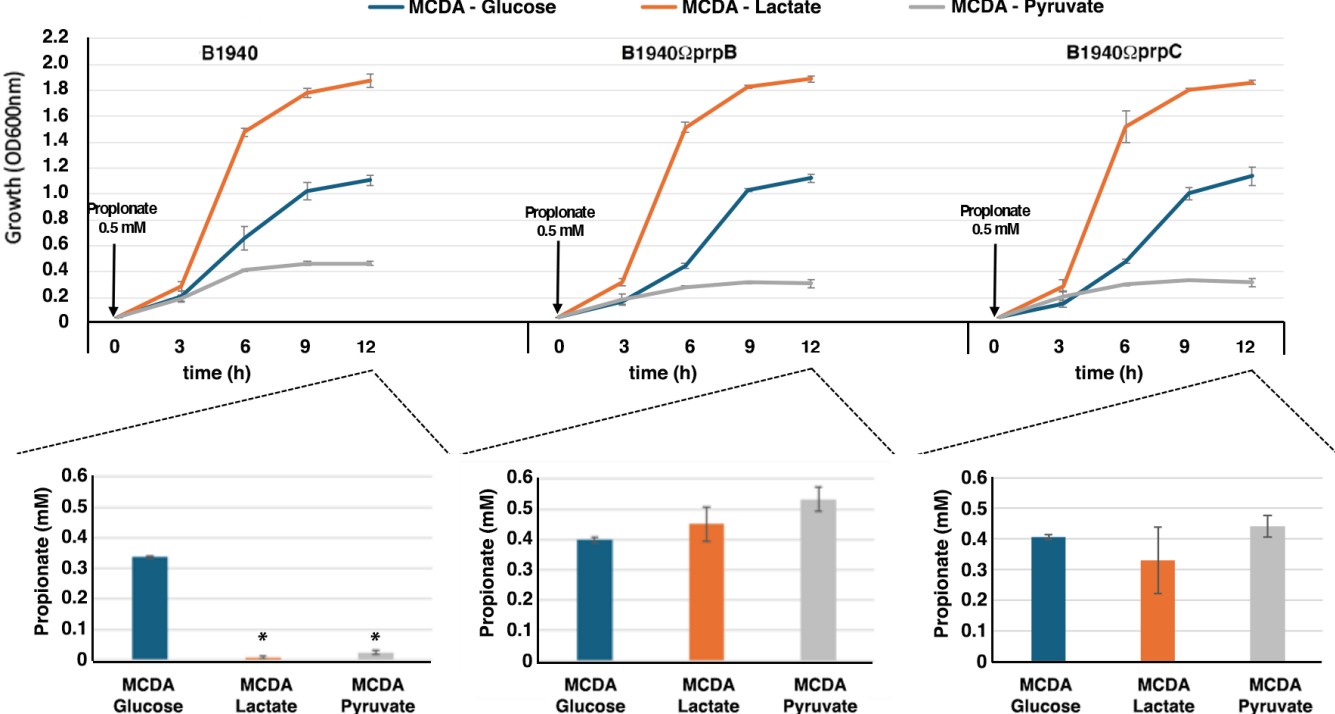

**FIG 6** Propionate utilization by *N. meningitidis* B1940 and *prpB*- and *prpC*-defective mutants. Extracellular propionate concentrations were determined by high-performance liquid chromatography after 12 h of growth of *N. meningitidis* B1940 and *prpB*- and *prpC*-defective mutants in MCDA-glucose, MCDA-pyruvate, or MCDA-lactate supplemented with 0.5 mM propionate. The data at each time point are the means and standard deviations of three independent experiments. Asterisks indicate statistically significant differences ($P < 0.05$) between MCDA-Lactate or MCD-Pyruvate and MCD-Glucose.

acetic acid. Noteworthy is that the *kbuT*-defective mutant tested positive for all these substrates except α-ketobutyric acid (Fig. 8A; File S6). This finding suggests that *kbuT* may encode a transporter required for the assimilation of α-ketobutyric acid, which can be directly transformed into propionyl-CoA, thus entering the MCC (Fig. 4). In support of this hypothesis, the *prp-B* and *prpC*-defective mutants of *N. meningitidis* showed a weaker positivity to α-ketobutyric acid than the isogenic wild-type B1940 strain, while in the other tests, they had the same behavior as the wild-type (Fig. 8A; File S6). As expected, *N. lactamica*, lacking the MCC, also tested negative for α-ketobutyric acid.

To further support this hypothesis, we tested the effects of α-ketobutyrate supplementation on meningococcal growth in MCDA-lactate (Fig. 8B; File S5). To this end, different amounts of α-ketobutyrate (0.25, 0.5, and 1 mM) were added to low cell density (0.1 $OD_{600 \, nm}$) cultures of the wild-type MC58 and *kbuT*-defective strain, and growth was monitored for the next 9 h. The addition of α-ketobutyrate resulted in a slowing of growth in the first 3 hours. This slowing was more marked for the *kbuT*-defective mutant than for the wild-type strain. At the highest tested concentration of α-ketobutyrate, 1 mM, growth of the *kbuT*-defective mutant was also slowed during the subsequent 3 h, whereas growth of the wild-type strain was similar to that of the control without α-ketobutyrate. This result demonstrated that the *kbuT*-defective mutant was more sensitive to α-ketobutyrate inhibition than the wild-type strain. We hypothesized that this could be due to the inability of the *kbuT*-defective mutant to take up and metabolize α-ketobutyrate. Thus, we tested the residual concentration of α-ketobutyrate after the growth of the wild-type strain and the *kbuT*-defective mutant in MCDA-lactate supplemented with 1 mM α-ketobutyrate. The results of gas chromatography-liquid chromatography-electrospray ionization-mass spectrometry (LC-ESI(-)-MS) analyses showed that almost the entire α-ketobutyrate supplement had been consumed by the wild-type

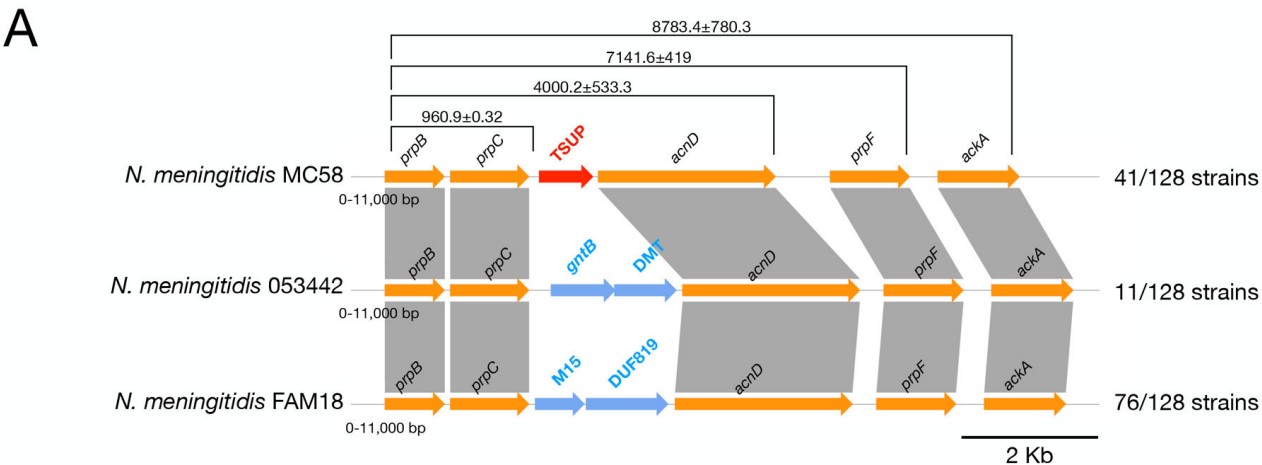

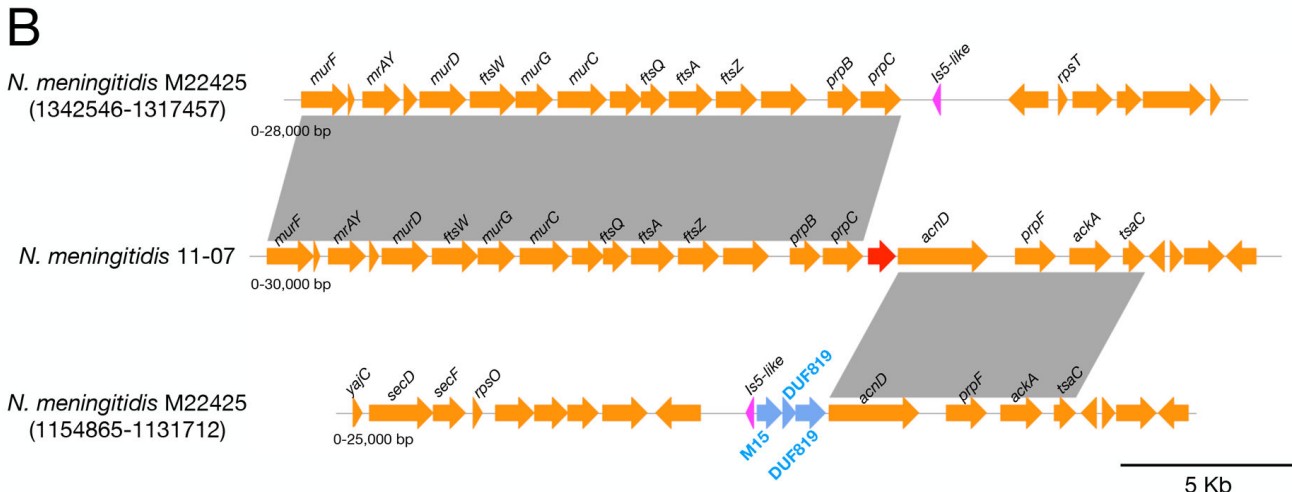

**FIG 7** Organization and variation of the MCC gene cluster locus of *N. meningitidis*. (A) Genetic map of the MCC gene cluster in 128 completely annotated genomes of *N. meningitidis* strains. The main differences concern the region between *prpC* and *acnD*, with three possible configurations linked to the genes located in this region coding for TSUP (*kbuT*), *gntB*-DMT, or M15-DUF819. Representative meningococcal type strains are indicated to the left of each map. (B) Rearrangement of the MCC gene cluster in *N. meningitidis* strain M22425.

strain after 9 h of growth, whereas approximately 80% remained in the culture medium of the *kbuT*-defective mutant (Fig. 8B).

## The MCC gene cluster in *N. meningitidis* and other *Neisseria* species

We studied the presence of the MCC gene cluster in pathogenic and commensal *Neisseria* species and found *prpB* and *prpC* in pathogenic *N. meningitidis* and *N. gonorrhoeae*, but not in the human-dwelling commensal species *N. lactamica* and *N. cinerea* (Fig. 9A and B). However, we noted that, while in *N. meningitidis,* the entire MCC gene cluster was present in 100% of the strains analyzed, in *N. gonorrhoeae,* this gene cluster is decaying (Fig. S9A). Indeed, in *N. gonorrhoeae,* only *prpC* was present in 100% of the strains analyzed, while *acnD* and *prpF* were pseudogenes in 100% of the strains, and, in about 25% of these, even for *prpB* and *kbuT,* only pseudogenes were present (Fig. S9B).

Notably, the MCC gene cluster was also found in several *Neisseria* species that colonize humans less frequently, *N. flavescens*, *N. subflava*, *N. macacae*, *N. sicca,* and *N. mucosa* (Fig. 9A and B; Fig. S10). However, in these bacteria, the gene cluster is located in different genomic regions than where it is found in *N. meningitidis* and *N. gonorrhoeae*. In particular, in *N. flavescens* and *N. subflava,* the cluster maps in a region between the

## A

| | N. lact. 995 | N. lact. 411 | N. mening. B1940 | N. mening. B1940 ΩprpB | N. mening. B1940 ΩprpC | N. mening. MC58 | N. mening. MC58 ΩkbuT | |
|---|---|---|---|---|---|---|---|---|
| p-Hydroxy- Phenylacetic Acid | 0,0 | 0,3 | 0,0 | 0,0 | 0,0 | 0,0 | 0,0 | |
| Methyl Pyruvate | 1,0 | 0,0 | 0,9 | 1,1 | 1,1 | 0,7 | 0,3 | |
| D-Lactic Acid Methyl Ester | 0,3 | 0,4 | 1,0 | 1,0 | 0,9 | 1,1 | 1,1 | |
| L-Lactic Acid | 1,2 | 1,1 | 1,0 | 1,1 | 1,0 | 1,0 | 1,0 | |
| Citric Acid | 0,0 | 0,0 | 0,2 | 0,2 | 0,0 | 0,0 | 0,0 | |
| α-Keto-Glutaric Acid | 0,0 | 1,0 | 1,1 | 1,1 | 1,0 | 1,1 | 0,9 | |
| D-Malic Acid | 0,0 | 0,0 | 0,0 | 0,0 | 0,0 | 0,0 | 0,0 | |
| L-Malic Acid | 0,0 | 1,0 | 1,0 | 1,0 | 1,1 | 1,0 | 1,0 | |
| Bromo-Succinic Acid | 0,0 | 0,3 | 0,8 | 1,0 | 0,8 | 0,6 | 0,6 | |
| Nalidixic Acid | 0,4 | 1,0 | 1,0 | 0,9 | 0,9 | 0,7 | 0,6 | |
| Lithium Chloride | 0,0 | 0,0 | 0,6 | 0,5 | 0,2 | 0,9 | 1,0 | |
| Potassium Tellurite | 0,0 | 0,0 | 0,0 | 0,0 | 0,0 | 0,1 | 0,0 | |
| Tween 40 | 0,0 | 0,0 | 0,0 | 0,0 | 0,0 | 0,0 | 0,0 | |
| γ-Amino-Butyric Acid | 0,0 | 0,0 | 0,0 | 0,0 | 0,0 | 0,0 | 0,0 | |
| α-Hydroxy-Butyric Acid | 0,2 | 1,0 | 1,0 | 1,1 | 0,9 | 1,0 | 1,0 | |
| β-Hydroxy-D,L-Butyric Acid | 0,0 | 0,0 | 0,0 | 0,0 | 0,0 | 0,0 | 0,0 | |
| α-Keto-Butyric Acid | 0,0 | 0,0 | 1,1 | 0,7 | 0,7 | 1,0 | 0,0 | * |
| Acetoacetic Acid | 0,0 | 0,4 | 0,7 | 1,1 | 0,8 | 0,5 | 0,5 | |
| Propionic Acid | 0,0 | 0,0 | 0,0 | 0,0 | 0,0 | 0,0 | 0,0 | |
| Acetic Acid | 0,0 | 0,3 | 0,9 | 1,0 | 1,0 | 1,1 | 0,7 | |
| Formic Acid | 0,0 | 0,0 | 0,0 | 0,0 | 0,0 | 0,0 | 0,1 | |
| Aztreonam | 0,0 | 0,2 | 0,6 | 0,9 | 0,8 | 0,8 | 1,1 | |
| Sodium Butyrate | 0,0 | 0,3 | 0,6 | 0,7 | 0,4 | 0,9 | 0,9 | |
| Sodium Bromate | 0,0 | 0,0 | 0,0 | 0,0 | 0,0 | 0,0 | 0,0 | |

## B

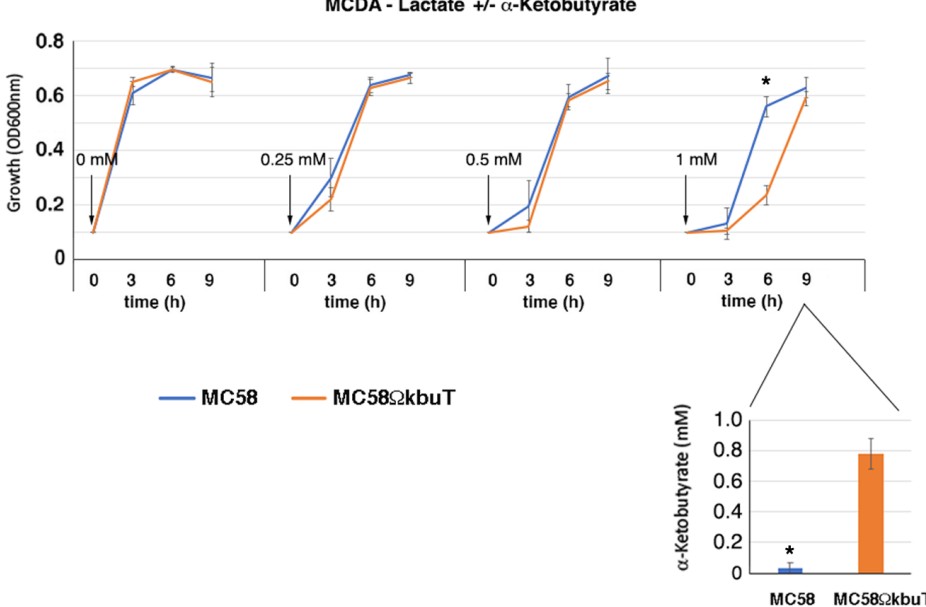

**FIG 8** Phenotyping of *N. meningitidis* strains with BIOLOG Phenotype MicroArray system and growth curves of *N. meningitidis* MC58 and isogenic *kbuT*-defective mutant in the presence of α-ketobutyrate. (A) Phenotyping of *N. meningitidis* with BIOLOG Phenotype MicroArray system. (B) Growth curves of *N. meningitidis* MC58 and its isogenic *kbuT*-defective mutant in MCDA-lactate supplemented with different amounts of α-ketobutyrate (0, 0.25, 0.5, and 1 mM) (top), and high-performance liquid chromatography assay of residual α-ketobutyrate in the growth medium (bottom). Two independent *kbuT*-defective mutants were tested, along with the wild-type B1940, in three independent experiments as reported in File S5. Means and standard deviations are shown at each time point. Asterisks indicate statistically significant differences ($P < 0.05$) between B1940 and the *kbuT*-defective mutant at the corresponding time points.

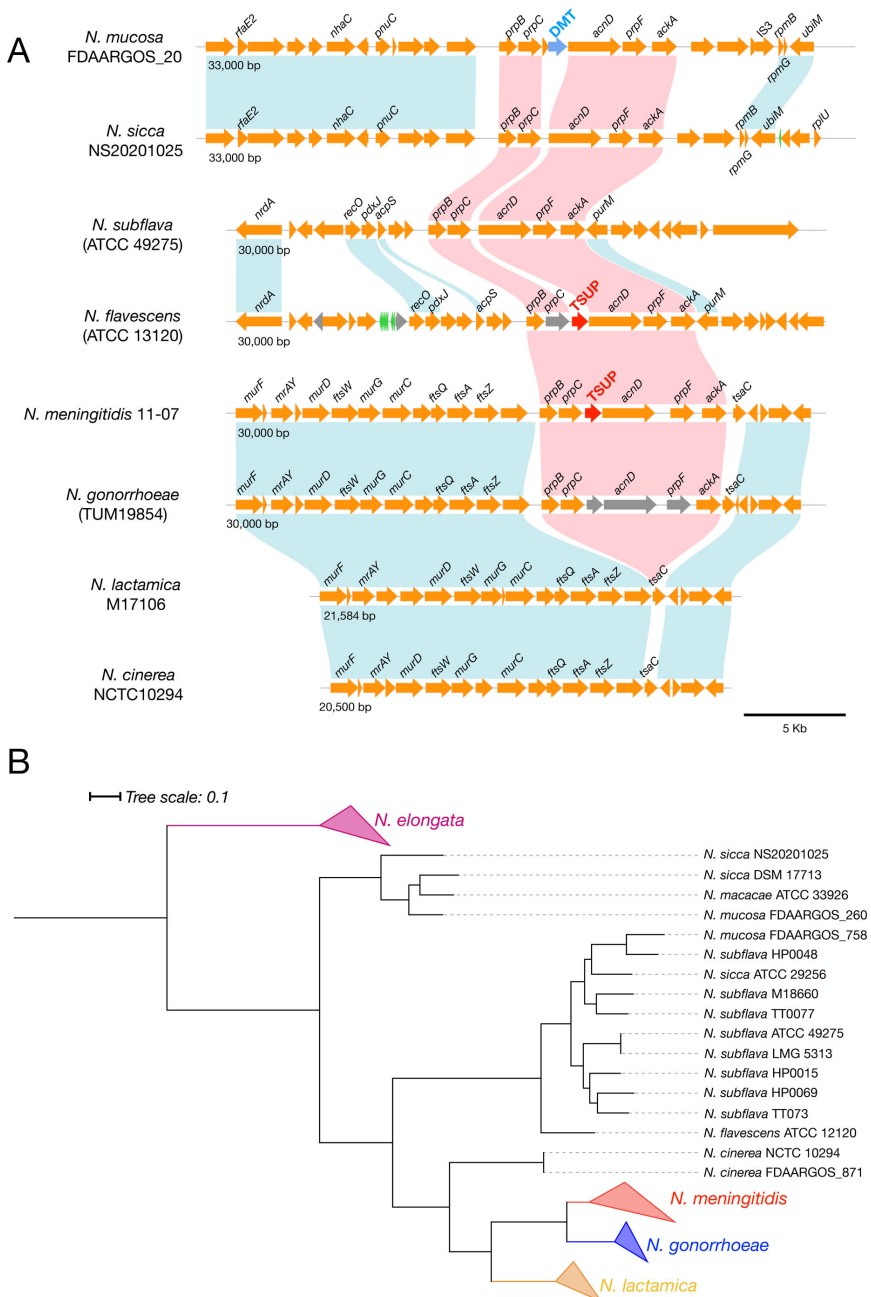

**FIG 9** Organization and variation of the MCC gene cluster locus in commensal and pathogenic *Neisseria* spp. (A) Genetic map of the MCC gene cluster in completely annotated genomes of commensal and pathogenic *Neisseria* spp. (B) Phylogenetic tree obtained by concatenating 16 genes and illustrating the relationships between pathogenic and commensal species belonging to the *Neisseria* genus.

*acpS* and *purM* genes, while in *N. sicca* and *N. mucosa,* it is located in a region between the *pnuC* and *rpmB* genes (Fig. 9A). In one strain of *N. sicca*, *N. mucosa,* and *N. macacae,* the gene encoding the DMT family transporter was located between *prpC* and *acnD* (Fig. S10).

As mentioned above, in *N. meningitidis,* the region between *prpC* and *acnD* shows intraspecies variation (Fig. 7A and B; Fig. 10), with three possible configurations: TSUP (*kbuT*), *gntB*-DMT, and M15-DUF819. Of note, the TSUP-encoding gene (*kbuT*) was found in many of the *N. meningitidis* lineages responsible for epidemics or sporadic outbreaks,

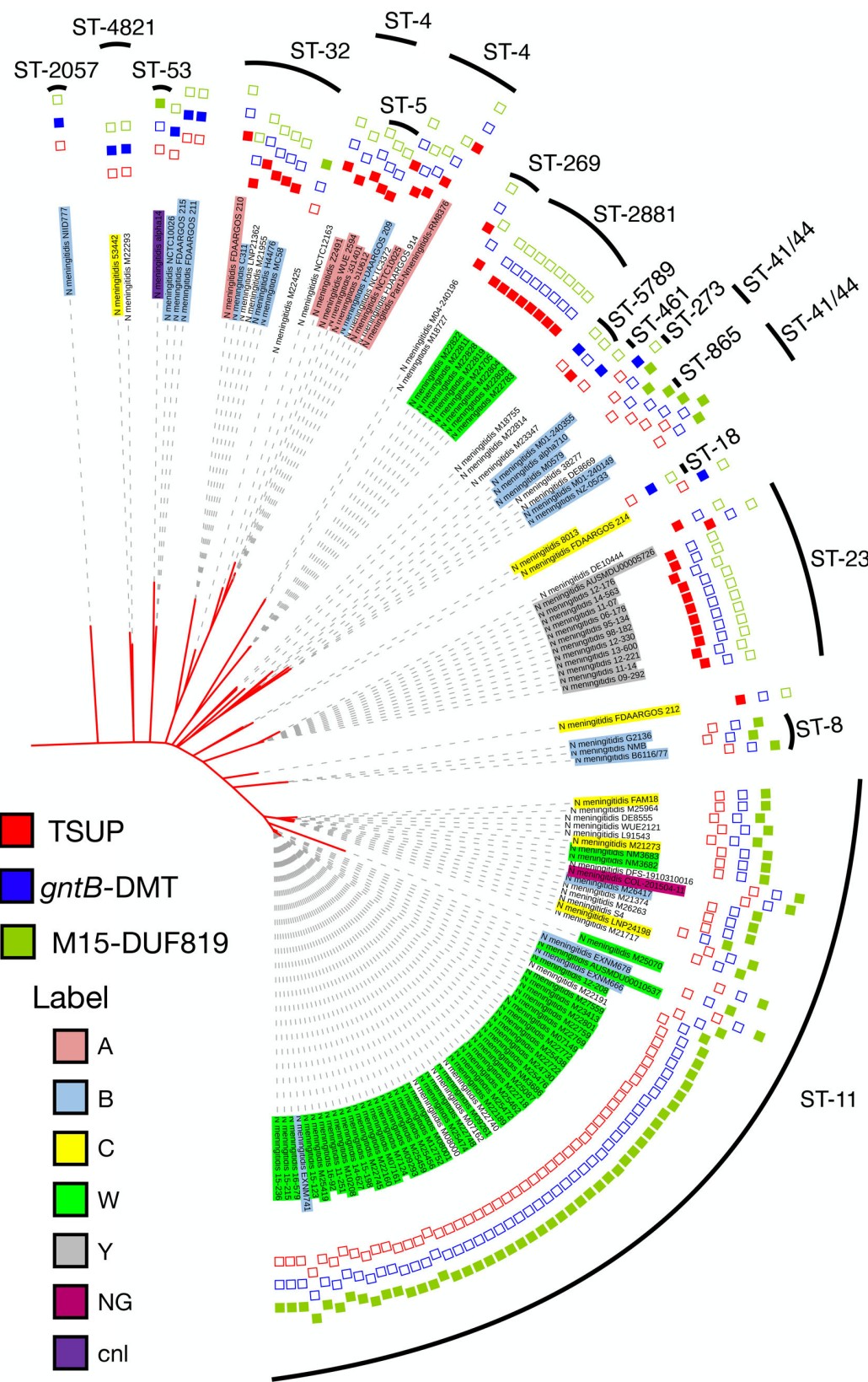

**FIG 10** Variation of the region between *prpC* and *acnD* of the MCC gene cluster locus in different *N. meningitidis* lineages. View of the tree branch containing strains belonging to the species *N. meningitidis*, highlighting serogroup, genotype, and the three possible configurations of the genetic region between *prpC* and *acnD* showing intraspecies variation: TSUP (*kbuT*), *gntB*-DMT, and M15-DUF819.

including strains belonging to the clonal complexes ST-32 (mostly of serogroup B), ST-4 (mostly of serogroup A), ST-2881 (serogroup W135), and ST-1655 (mostly of serogroup Y), while in the hypervirulent clonal complex, the ST-11 (mostly of serogroup W135, and other of serogroup C) *kbuT* was replaced by the M15- and DUF819-encoding genes (Fig. 10).

## DISCUSSION

It has been argued that the MCC gene cluster confers a selective advantage to *N. meningitidis* over *N. lactamica* in the human nasopharynx of young adults, where the colonization of propionic bacteria increases over time (34). Indeed, the MCC allows attenuation of the toxicity of the propionate produced by these bacteria. This belief, however, is partially inconsistent with evidence that the meningococcus cultured with lactate, a major carbon source in the human nasopharynx for commensal and pathogenic *Neisseria* spp., did not show growth inhibition in the presence of high concentrations of propionate in the MCDA minimal medium, even when the MCC was nonfunctional, such as in *prpB*- and *prpC*-defective mutants (Fig. 1). Instead, in line with previous findings (34, 59), our results confirm that the MCC confers to the meningococcus the ability to use propionate as an additional carbon source during late stages of growth on glucose, a carbohydrate that at high concentrations represses the MCC gene cluster (19), or growth on suboptimal carbon sources, such as pyruvate (Fig. 1 and 4). These considerations suggest that the role of MCC is to assimilate rather than detoxify propionate.

The different sensitivity to propionate of meningococci grown on glucose, pyruvate, and lactate led us to investigate the possible mechanisms of growth inhibition by propionate, a topic widely addressed in various microbial systems (38). Since propionate toxicity was dramatically reduced in the *ackA2* mutant during both growth on pyruvate and lactate (Fig. 3A), this toxicity is presumed to be due to specific mechanisms caused by the accumulation of propionyl-CoA and its detrimental effects on bacterial bioenergetics (38). Among these mechanisms, the one most involved in both the bacterial and fungal systems is the inhibition of pyruvate dehydrogenase (40, 41).

Unexpectedly, propionate toxicity was not reduced in the *ackA2* mutant during the growth on glucose (Fig. 3A), and this could be attributed to the fact that *ackA2* is significantly downregulated when meningococci are cultured with glucose as the main carbon source (19). We hypothesized that under these conditions, propionyl-CoA biosynthesis is primarily carried out by the acetate kinase/propionate kinase encoded by the *ackA1*, consistent with the results of reverse transcriptase real-time PCR experiments (Fig. 3B). Indeed, we found that an *ackA1*-defective mutant grew in MCDA-glucose containing 5 mM propionate (Fig. 3A).

As mentioned above, we found that the growth of *prpB* and *prpC* mutants on lactate was not inhibited by propionate even at high concentrations and that the wild-type was more sensitive than the mutants (Fig. 1). We initially hypothesized that the higher resistance of bacteria to propionate when grown on lactate compared to glucose or pyruvate could be due to a mechanism of competition with lactate in the entry of propionate into the bacteria, but this hypothesis was ruled out based on the available evidence (Fig. 5 and 6). Therefore, for some still unclear reason, bacteria growing on lactate appear less sensitive than those growing on glucose or pyruvate to the inhibitory effect of propionate on bacterial growth.

Particularly intriguing is the different sensitivity of bacteria grown on lactate compared to those grown on pyruvate, since a different distribution of the intracellular carbon flux cannot even be questioned, as is the case for bacteria grown on glucose compared to those grown on lactate or pyruvate that require sustained gluconeogenic activity. One possible explanation is that lactate is a very favorable carbon source for meningococci, which possess very efficient lactate uptake and rapidly metabolize lactate to pyruvate (Fig. 4), feeding the respiratory chain (3, 60). Therefore, growth on lactate

could attenuate the detrimental effects of propionate on pyruvate dehydrogenase and, in general, on bacterial bioenergetics.

We do not know the exact mechanism, but we hypothesize that allosteric regulation of pyruvate dehydrogenase by pyruvate may be involved. Indeed, the pyruvate dehydrogenase complex is allosterically stimulated by pyruvate in *E. coli* (61), *Azotobacter vinelandii* (62), and presumably other Gram-negative bacteria (63). If this were also true for the pyruvate dehydrogenase complex of *N. meningitidis*, it could be hypothesized that, when meningococci are grown in MCDA-lactate, this carbon source is readily converted to pyruvate, which allosterically stimulates the pyruvate dehydrogenase complex, which, under these conditions, may be less sensitive to inhibition by propionyl-CoA. Conversely, in MCDA-pyruvate, pyruvate uptake may be inefficient as indicated by poorer growth in MCDA-pyruvate compared to MCDA-lactate, pyruvate concentration is low, and pyruvate dehydrogenase is poorly allosterically stimulated. Consequently, it may be highly sensitive to propionyl-CoA inhibition. In MCDA-glucose, glucose uptake is not as efficient as lactate as the main carbon source (meningococcal growth in MCDA-glucose is lower than in MCDA-lactate). Furthermore, glucose must enter the Entner-Doudoroff pathway to be metabolized to pyruvate. As a result, the intracellular concentration of pyruvate is expected to be lower than in bacteria growing in MCDA-lactate, and the pyruvate dehydrogenase is allosterically stimulated less than in bacteria growing in MCDA-lactate. As a result, it may be more sensitive to inhibition by propionyl-CoA. This hypothesis is supported by the evidence that intracellular pyruvate levels in mid-log growing bacteria are different in MCDA-glucose, MCDA-lactate, and MCDA-pyruvate, with pyruvate levels particularly low in MCDA-pyruvate (Fig. 6SC). It can also be noted that in MCDA-pyruvate without propionate, both the *ackA1*- and the *ackA2*-defective mutants grew faster than the wild-type strain (Fig. 3A). Overall, these results are consistent with inefficient utilization of this carbon source, particularly in the presence of acetate kinase activity, which can direct the carbon source toward acetate (Fig. 4).

Regarding pyruvate uptake, it is also noteworthy that, unlike the *lctP*-defective mutant of *N. gonorrhoeae* that was unable to grow in GC-broth containing pyruvate in place of glucose (64), the *lctP*-defective mutant of *N. meningitidis* did not show a growth defect in MCDA-pyruvate, suggesting differences in pyruvate uptake systems and substrate affinity of the LctP transporter between gonococcus and meningococcus.

One main finding of the present study is that in the genome of 41 out of 128 meningococcal strains, there is a gene, called *kbuT*, encoding an unknown transporter belonging to the 4-TSUP family, located between *prpC* and *acnD* of the MCC gene cluster (Fig. 7, 9 and 10). Genetic inactivation of *kbuT* did not alter sensitivity to propionate (Fig. S8C; File S5), but led to the inability, in a Phenotype MicroArray system, to oxidate α-ketobutyric acid (Fig. 8A; File S6), an α-keto acid that can be used as a carbon source via the MCC (Fig. 4). In contrast, the *kbuT*-defective mutant retained the ability to oxidate α-ketoglutaric acid, the other α-keto acid used by *N. meningitidis*. These results suggested that the MCC gene cluster has evolved in several *N. meningitidis* strains to utilize α-ketobutyric acid, in addition to propionic acid. This hypothesis was supported by the results of experiments on MCDA-lactate supplemented with α-ketobutyrate, which demonstrated that 1 mM α-ketobutyrate was assimilated and was able to stimulate the growth of the wild-type strain but not of the *kbuT*-defective mutant (Fig. 8B; File S5).

In the human host, α-ketobutyric acid is a degradation product of threonine via the activity of threonine/serine dehydratase and can be produced via the lysis of cystathionine (by cystathionine gamma lyase) in the methionine and homocysteine degradation pathway. In human cells, α-ketobutyric acid can be converted to propionyl-CoA and then to methylmalonyl-CoA, which can be converted to succinyl-CoA and thus enter the TCA cycle. In *N. meningitidis*, as in many other bacteria, α-ketobutyric acid is a precursor in the biosynthesis of isoleucine, an amino acid that is nutritionally essential for humans. Additionally, meningococci, which lack the methylmalonyl-CoA pathway, can convert

α-ketobutyric acid to propionyl-CoA, which can be utilized through the MCC which, in turn, feeds the TCA cycle (Fig. 4).

The conversion of α-ketobutyrate to propionyl-CoA can be catalyzed by several enzymes: in human cells, this reaction can be catalyzed by branched-chain keto acid dehydrogenase; in *E. coli* and other Enterobacteriaceae, the reaction can be catalyzed by the main pyruvate formate lyase (PflB) or, under anaerobic conditions in which the threonine degradation pathway is induced, by TdcE, an enzyme that has both pyruvate formate lyase and 2-ketobutyrate formate lyase activity (65). In other microorganisms lacking these enzymes, such as *N. meningitidis*, conversion of α-ketobutyrate to propionyl-CoA can be catalyzed by the pyruvate dehydrogenase, although with a low catalytic efficiency (66–69).

In the human body, α-ketobutyric acid can reach high concentrations, especially in the intracellular environment. In particular, quantitative determination of free intracellular α-keto acids in a pooled sample of human polymorphonuclear neutrophils (PMNs) by reversed-phase fluorescence high-performance liquid chromatography (HPLC) revealed the following levels expressed as $10^{-2}$ fmol/cell: $1.31 \pm 0.70$ α-ketoglutarate; $6.41 \pm 3.95$ pyruvate; $4.05 \pm 3.95$ α-ketobutyrate; $2.37 \pm 1.24$ α-ketoisovalerate; $0.37 \pm 0.23$ α-ketoisocaproate; and $0.89 \pm 0.67$ α-keto-β-methylvalerate (70). Thus, among the α-keto acids tested, α-ketobutyrate is the most abundant, after pyruvate, in the intracellular environment of PMNs. Also in human serum, the concentration (expressed in µmol/L) of α-ketobutyrate, determined with the same method, was rather high ($8.36 \pm 1.81$), similar to that of α-ketoglutarate ($7.34 \pm 2.05$), but clearly lower than that of pyruvate ($77.50 \pm 28.90$) and of the other α-keto acids analyzed (70). This suggests that, especially in the intracellular environment, *N. meningitidis* can utilize α-ketobutyrate as a carbon source, via the MCC cycle, to fuel the TCA cycle.

Indeed, although *N. meningitidis* is traditionally considered an extracellular pathogen, during its infectious cycle, it must necessarily cross cellular barriers to cause the invasive meningococcal disease. To cross the nasopharyngeal and the blood-brain barrier, it establishes a complex cross-talk with epithelial and endothelial cells, respectively, which is based on complex metabolic interactions (3). In addition, *N. meningitidis* can avoid killing by PMNs, and this was also shown to be dependent on its metabolic capability, such as the ability of the bacterium to acquire L-glutamate and convert it to glutathione (32). More importantly, *N. meningitidis* can escape the internalization vacuole in infected cells and reach the cytosol (71–73), and therefore easily accesses α-ketobutyrate.

Interestingly, pangenome analysis of *N. gonorrhoeae* suggests that the MCC genes are in decay in this species (Fig. S9). This finding may reflect the different lifestyles and diseases caused by *N. gonorrhoeae* and *N. meningitidis*. *N. meningitidis*, a frequent asymptomatic colonizer of the human upper respiratory tract, can cause devastating and fulminant infections of the blood and brain in susceptible individuals. *N. gonorrhoeae*, a sexually transmitted pathogen, primarily infects the urogenital tract, causing intense local inflammation and other clinical manifestations and has a tendency to establish chronic infection (74). Unlike the meningococcus, which has the ability to exit the internalization vacuole to escape lysosomes and rapidly spread, the gonococcus establishes its intracellular niche within a vacuole, the gonococcus-containing vacuole (GCV) (75).

## Conclusion

In this study, we provide new elements to decode the functionality of the methylcitrate pathway in *N. meningitidis*. This pathway allows the propionic acid assimilation as a supplemental carbon source, mitigating its toxicity in colonized host microenvironments, where propionic acid may be abundant. Specifically, using isogenic mutants, we found that the growth inhibition by propionic acid changes drastically depending on the main carbon source used by the bacterium. This finding prompted us to explore alternative functions of the MCC. In particular, we investigated a previously unexplored intraspecies variant of the MCC gene cluster in *N. meningitidis*. A putative α-ketobutyric acid

transporter encoding gene has been phylogenetically and functionally characterized, extending the role of the MCC to the utilization of additional carbon sources in host microenvironments relevant to meningococcal infection.

## MATERIALS AND METHODS

### Bacterial strains and growth conditions

The wild-type neisserial strains used in this study were *N. meningitidis* B1940 (B:NT:P1.3,6,15; lipooligosaccharide [LOS] immunotype: L3,7,9) (76), *N. meningitidis* MC58 (B.15.P1.7,16; LOS immunotype: L3), and two isolates of *N. lactamica*, NL995 and NL411, from human pharynx (77).

The solid medium used for neisserial strains growth was GC agar base (OXOID), supplemented with 1% (vol/vol) Polyvitox (OXOID) and, when required, with erythromycin (7 µg/mL), at 37°C in a 5% $CO_2$ incubator. The liquid medium used for growth was GC broth, whose composition (per liter) is as follows: 15 g proteose peptone, 0.5 g corn starch, 4 g $K_2HPO_4$, 1 g $KH_2PO_4$, 5 g NaCl, 1% (vol/vol) Polyvitox (OXOID). Neisserial strains were also cultured in the CDM MCDA at 37°C in a shaking incubator at 200 rpm (6, 24). The final concentration of each component in MCDA is listed: 100 mM NaCl, 2.5 mM KCl, 7.5 mM $NH_4Cl$, 7.5 mM $Na_2HPO_4$, 1.25 mM $KH_2PO_4$, 2.2 mM $Na_3C_6H_5O_7 \cdot 2H_2O$, 2.5 mM $MgSO_4 \cdot 7H_2O$, 0.0075 mM $MnSO_4 \cdot H_2O$, 8.0 mM L-glutamic acid, 0.5 mM L-arginine, 2.0 mM glycine, 0.2 mM L-serine, 0.06 mM L-cysteine·HCl·$H_2O$, 0.5% (vol/vol) glycerin, 0.25 mM $CaCl_2 \cdot 2H2O$, and 0.01 mM $Fe_2(SO_4)_3$. When required, the MCDA medium was supplemented with glucose, sodium lactate, sodium pyruvate, sodium propionate, and sodium α-ketobutyrate at the indicated concentrations. All reagents were purchased from Sigma-Aldrich.

*Escherichia coli* strain DH5α (F- Φ80d *lacZ*△M15 *endA1 recA1 hsdR17 supE44 thi-1* λ⁻ *gyrA96* △[*lacZYA-argF*] *U169*) was used in cloning procedures. This strain was grown in Luria–Bertani (LB) medium. To allow plasmid selection, LB medium was supplemented with ampicillin (75 µg/mL) or chloramphenicol (25 µg/mL).

### DNA procedures and construction of plasmids and recombinant strains

Chromosomal and plasmid DNA were obtained by standard methods (78).

Oligonucleotides used in this study as primers in PCRs are listed in Table 1. The amplification reactions were performed under the following conditions for 35 cycles: 94°C for 45 s, 65°C for 45 s, and 72°C for 45 s. Genomic DNA from strains B1940 and MC58 was used as templates.

The *Neisseria-E. coli* shuttle vector pDEX (24) was used to construct B1940ΩprpB, B1940ΩprpC, B1940ΩackA1, B1940ΩackA2, B1940ΩlctP, and MC58ΩkbuT. DNA corresponding to the central segment of the *prpB* (NMB430 in strain MC58), *prpC* (NMB431 in strain MC58), *ackA1* (NMB1518 in strain MC58), *ackA2* (NMB435 in strain MC58), *lctP* (NMB0543 in strain MC58), and *kbuT* (NMB0432 in strain MC58) genes were amplified using the following primers, respectively: PrpBF-PrpBR; PrpCF-PrpCR; AckA1F-AckA1R; AckA2F-AckA2R; LctPF-LctPR; and KbuTF-KbuTR (Table 1). The resulting BamH1-restricted PCR products (amplimer lengths: 546 bp, 737 bp, 766 bp, 662 bp, 535 bp, and 570 bp, respectively) were ligated into the BamHI site of pDEX, harboring a functional erythromycin-resistance cassette, obtaining the following constructs, respectively: pDE*DprpB*, pDE*DprpC*, pDE*DackA1*, pDE*DackA2*, pDE*DlctP*, and pDE*DkbuT*.

The constructs pDE*DprpB*, pDE*DprpC*, pDE*DackA1*, pDE*DackA2*, and pDE*DlctP* were used to transform *N. meningitidis* B1940, the construct pDE*DkbuT* was used to transform *N. meningitidis* MC58. Transformations were performed using 0.1 to 1 µg of plasmid DNA. Transformants were selected on GC agar medium supplemented with erythromycin (7 µg/mL).

Successful gene inactivation was demonstrated by PCR analysis using the following knockout-specific primer pairs: PrpBKOF and PrpBKOR for B1940WprpB, PrpCKOF and

**TABLE 1** Oligonucleotides used in this study

| Name | Sequence |
|---|---|
| PrpCF: | 5′-gcctgcggatccgtgtgattaaagttttggaaag-3′ |
| PrpCR: | 5′-ttggtaggatccggcagagaaccagtccagattc-3′ |
| PrpCKOF: | 5′-ctggatttggcacaaaaatgcgag-3′ |
| PrpCKOR: | 5′-CTCAAGAACGTGTGCGCTCCAACC-3′ |
| RT-PrpCF: | 5′-ccatgcacgtttcactgattc-3′ |
| RT-PrpCR: | 5′-gatcacgcgggcggtaaaggtag-3′ |
| PrpBF: | 5′-CAAAGCGGATCCAAAGCCATCTATCTGTCCGGC-3′ |
| PrpBR: | 5′-cggagtggatccaaactcggtaatgttcgccaac-3′ |
| PrpBKOF: | 5′-ggttgcgtcaatgcttattttgcac-3′ |
| PrpBKOR: | 5′-ggcacgggtttgcatactgtccac-3′ |
| RT-PrpBF: | 5′-CGCTGCGGCCACCGTCCGAAC-3′ |
| RT-PrpBR: | 5′-gttctcatcaacgcgcg-3′ |
| prpCompF: | 5′-CTATGGTCTAGATTGTTGCCGTATTGTGGGCACTG-3′ |
| prpCompR: | 5′-GTTCTCCTCTAGATCTGTTTTTCTTGTGGTTTGAGG-3′ |
| PrpB_CompF: | 5′-GGAGAAATATGATGAGTCAACAC-3′ |
| PrpB_CompR2: | 5′-GATGCGGATTTGTTGAAAGGCAG-3′ |
| PrpC_CompF2: | 5′-CTGAAAACCCGAAACCCATAAAAAC-3′ |
| PrpC_CompR2: | 5′-CAAGGTCGTCTGAAATCCTTAGC-3′ |
| KbuTF: | 5′-cctcccggatccgcaattgccaccaacaagctg-3′ |
| KbuTR: | 5′-CAGCTTGGATCCGAAGCGGACGGCAAATCTCGC-3′ |
| KbuTKOF: | 5′-gatgattgccggatttatcgatgc-3′ |
| KbuTKOR: | 5′-ctgatacagcggatttctctcgtc-3′ |
| AckA1F: | 5′-GGTCCTGGATCCCGGCAGCGGCGAAGTCCTGCTC-3′ |
| AckA1R: | 5′-GCCGAGGATCCCGGATTTTTTGTTCAGCATTTCAG-3′ |
| AckA1KOF: | 5′-GAACTGCGGCAGCTCGTCCCTC-3′ |
| AckA1KOR: | 5′-CGCCATACTGCCGATGTATTTG-3′ |
| RT-AcKA1F: | 5′-cgtcagcggcggcgaactgtac-3′ |
| RT-AcKA1R: | 5′-ggttgtgcaggggggcgag-3′ |
| AckA2F: | 5′-CATCAGCGGGATCCTTGCCGCACAGGAACATTTC-3′ |
| AckA2R: | 5′-CAAGATAGGATCCGGTTTTGGCACGGATATTAC-3′ |
| AckA2KOF: | 5′-GCACGACCGCATCAAAGCCATCGG-3′ |
| AckA2KOR: | 5′-GCAATCATCAGTTCTTCATTGGTC-3′ |
| RT-AcKA2F: | 5′-CTGCCGCACCCTCGAAATCGC-3′ |
| RT-AcKA2R: | 5′-GTATTTGGCGAGGCGGTAGG-3′ |
| LctPF: | 5′-CTTCGCAGGATCCGTCATCCCCGTCATCGGCTTG-3′ |
| LctPR: | 5′-CAGCAGGATCCAAATCCAAACGGTCAGCACAAAC-3′ |
| LctPKOF: | 5′-CTGAACCTGAGTGCCGAAGACATC-3′ |
| LctPKOR: | 5′-GGTTACGGAATAAATCGTCCAGG-3′ |
| 16Suniv-1: | 5′-CAGCAGCCGCGGTAATAC-3′ |
| 16 S-r: | 5′-CTACGCATTTCACTGCTACACG-3′ |
| pIAF: | 5′-GCTTGCATGCCTGCAGGTCGAC-3′ |
| ErmR: | 5′-GTTCATATTTATCAGAGCTCGTG-3′ |

PrpCKOR for B1940ΩprpC, AckA1KOF and AckA1KOR for B1940WackA1, AckA2KOF and AckA2KOR for B1940WackA2, LctPKOF and LctPKOR for B1940WlctP, and KbuTKOR and KbuTKOF for MC58ΩkbuT. Each gene-specific primer was challenged with pDEX-specific primers, pIAF and/or ErmR, to confirm the insertional inactivation of the targeted gene.

B1940ΩprpB and B1940ΩprpC were genetically complemented by transformation with the integrative plasmid pNLCT-*prpBC*. To construct this plasmid, the *Neisseria-E. coli* shuttle vectors pNLCT1 were used (79). pNLCT1 is a pACYC184 derivative containing the following: (i) the chloramphenicol resistance genetic determinant Cm$^r$, as a selectable marker; (ii) a chromosomal region necessary for its integration into the *Neisseria* chromosome (79); (iii) a specific DNA uptake sequence which is required for

natural transformation; and (iv) several unique restriction sites that are useful for cloning. pNLCT1 integrates into the *leuS* region of the *N. meningitidis* chromosome at high frequencies by transformation-mediated recombination. The genomic region spanning from the promoter element located upstream of the *prpB* gene to the region upstream of the *acnD* gene was amplified from the B1940 strain, using the primers prpCompF and prpCompR (Table 1).

Successful genetic complementation was demonstrated by PCR analysis using the following oligonucleotides: PrpB_CompF-PrpB_CompR2 and PrpC_CompF2-PrpC_CompR2, specific for the *prpB* and *prpC* full-length genes, respectively.

## Real-time reverse transcriptase PCR

*N. meningitidis* B1940 was grown at 37°C with shaking to early logarithmic phase ($OD_{600 nm}$ = 0.4) and to late logarithmic phase ($OD_{600 nm}$ = 0.8) in MCDA-glucose, MCDA-lactate, and MACDA-pyruvate. Total RNA was extracted from bacteria, quantified using Nanodrop (Thermo Scientific), and 1 µg of each RNA was reverse transcribed using random primers (2.5 µM) with SuperScript III Reverse Transcriptase (Invitrogen). Real-time PCR was performed on a CFX96 System (Bio-Rad) with SsoAdvanced Universal SYBR Green Supermix (Bio-Rad) and the following primers, listed in Table 1: RT-PrpCF/RT-PrpCR for *prpC* mRNA levels, RT-PrpBF/RT-PrpBR for *prpB* mRNA levels, RT-AcKA2F/RT-AcKA2R for *ackA2* mRNA levels, RT-AcKA1F/RT-AcKA1R for *ackA1* mRNA levels, 16Suniv-1/16S-r for 16S rRNA levels. The real-time PCR protocol used was: 120 s at 95°C for one cycle, and 30 s at 94°C, 30 s at 55°C, 30 s at 72°C for 35 cycles. Real-time PCRs were run in triplicate, and data were normalized to the RNA levels of 16S rRNA, and relative mRNA expression was calculated using the ΔΔCt method.

## Assay of extracellular propionate and α-ketobutyrate concentrations

Gas chromatography and LC-ESI(-)-MS were used to determine the residual, extracellular propionate and α-ketobutyrate concentration, respectively, in the MCDA growth medium, supplemented with different carbon sources, as indicated. Briefly, one milliliter of the cultures incubated at 37°C in a shaking incubator at 200 rpm was collected after 12 h of growth and centrifuged at 12,000 × *g* for 5 min. The supernatant was filtered through a 0.2 µm pore size filter and stored at −80°C until the analyses were performed. For propionate determination by gas chromatography, each sample was mixed 1:1 with 0.13 M potassium phosphate buffer at pH 3. One microliter of the freshly acidified samples was injected in splitless mode in an Agilent 7890 GC-FID (Agilent Technologies). Propionate was separated on a polyethylene glycol (PEG)-type GC column (J&W DB-FATWAX Ultra Inert, 30 m × 0.25 mm × 0.25 µm) using helium as carrier gas at a constant flow of 1.5 mL/min and an isocratic oven program (130°C). Propionate was identified by its retention time using an authentic standard (Sigma-Aldrich srl, Milano), and its quantification was carried out by external calibration. α-Ketobutyrate concentration was analyzed by LC-ESI(-)-MS employing an Agilent 1100 HPLC on line with a LCQ DECA XP Plus (Thermo Finnigan) Ion Trap mass spectrometer equipped with an ESI source. Separations were performed on an analytical column Agilent InfinityLab Poroshell 120 EC-C18 (150 × 3.0 mm, 2.7 µm particle size) following linear gradient: A (water, 0.1% formic acid) 95% for 1 min, then B (acetonitrile, 0.1% formic acid) is increased to 95% in 20 min and kept at 95% for 5 min at flow rate of 0.3 mL/min. The injection volume was 5 µL. The mass spectrometer was operated in ESI (-) mode, and spectra were acquired in the range from *m/z* 75 to 1,125. α-Ketobutyrate was identified by its retention time and mass spectrum using an authentic standard (Sigma-Aldrich srl, Milano), and its quantification was carried out by external calibration using the extracted ion chromatograms (*m/z* 101.02).

## Measurement of the intracellular pyruvate pool

To analyze the intracellular pyruvate pool, bacteria were grown in MCDA-glucose, MCDA-lactate, and MCDA-pyruvate to mid-log phase ($OD_{600nm}$ = 0.6); then, about $10^9$

cells were washed twice in $H_2O$ and centrifuged at $1,000 \times g$ for 10 min. Next, three volumes of 5% sulfosalicylic acid were added to the cells, which were treated with lysozyme (1 mg/mL), freeze-thawed twice in liquid nitrogen, then left for 5 min at 4°C. Finally, samples were centrifuged at $13,000 \times g$ for 10 min and pyruvate determination was carried out as follows. To each extract and Na-pyruvate standard, 10 μL of $d_4$-succinic acid (0.1 mg/mL) was added as an internal standard. After drying using a centrifugal evaporator (Labconco CentriVap) at controlled temperature and pressure (30°C and 20 mbar), the samples were methoximated at 40°C for 1 h with 20 μL of methoxyamine hydrochloride (20 mg/mL in anhydrous pyridine). The silylation was carried out with 40 μL of pure N-tert-Butyldimethylsilyl-N-methyltrifluoroacetamide (CAS 77377-52-7) at 110°C for 1 h. 1 μL of the derivatized samples was immediately injected in splitless mode in an Agilent GC/MS apparatus (7890B gas chromatograph 7683B autosampler and 5977 MSD, all from Agilent Technologies) and analytes separated on a 15 m (ID 320 μm, phase thickness 0.1 μm) DB5-HT (J&W Scientific, Folsom, CA) column. The injector temperature is set to 270°C with a purge vent time of 30 s. The flow rate through the column was 3 mL/min, with a three-step temperature program starting with an initial isocratic temperature of 50°C kept for 5 min, followed by a gradient rate of 10°C/min reaching a final temperature of 320°C kept for a duration of 5 min. The mass spectra were recorded between 50 and 750 m/z.

## Assay of intracellular glutathione levels

To analyze the intracellular glutathione pool, bacterial strains B1940, B1940ΩprpB, and B1940ΩprpC were grown in MCDA-glucose and MCDA-lactate for 4 h with shaking at 37°C. Then, 5 mM propionate was added to half the volume of the cultures, incubation was continued for 1 h, and optical density at 600 nm ($OD_{600nm}$) of the cultures was determined. Bacterial cells were washed twice in $H_2O$, centrifuged at $1,000 \times g$ for 10 min. Next, three volumes of 5% sulfosalicylic acid were added to the cells, which were treated with lysozyme (1 mg/mL), freeze-thawed twice in liquid nitrogen, then left for 5 min at 4°C. Finally, samples were centrifuged at $13,000 \times g$ for 10 min, and glutathione concentrations were measured in the supernatants with the Glutathione Assay Kit (Sigma), according to the manufacturer's instructions.

## Comparative genomic and phylogenetic analysis

Comparative genomics analysis was performed on 250 complete genomic sequences downloaded from NCBI-Genome, selecting only complete assemblies (Table S1). Of these sequences, 128 belonged to *N. meningitidis*, 97 to *N. gonorrhoeae*, and the remaining 25 to other *Neisseria* species (including eight from *N. subflava*, four from *N. lactamica* and *N. elongata*, three from *N. sicca*, two from *N. mucosa* and *N. cinerea*, and one from *N. flavescens* and *N. macacae*).

The protein sequences encoded by the genes *prpB* (WP_002224930.1), *prpC* (WP_002216540.1), TSUP family transporter (*kbuT,* WP_002216541.1), *acnD* (WP_002224931.1), *prpF* (WP_002224932.1), and *ackA* (WP_002222042.1) from *N. meningitidis* MC58 (NC_003112.2) were used as initial inputs. Protein sequences were analyzed using Biopython, specifically the pairwise2.align.globalms function, with a custom scoring system (+1 for a match, −1 for a mismatch, −0.5 for gap opening, and −0.1 for gap extension). The obtained scores were normalized based on protein length, with a threshold value set at 0.6. Subsequently, each genome was manually analyzed to identify the presence of any pseudogenes. Genetic maps were generated using pyGenomeViz (80).

Phylogenetic analysis was conducted by concatenating the nucleotide sequences of 15 protein-coding genes known to be vertically transmitted: *gatB, if2, miaB, hisS, gatA, fusA,* and nine genes encoding ribosomal proteins (*rplA, rplB, rplF, rplN, rplW, rpsH, rpsI, rpsK,* and *rpsM*) (81). Additionally, the gene encoding 16S rRNA was included.

Gene sequences were aligned using the MUSCLE algorithm in MEGA11 (82). MEGA11 was also used for gene concatenation and phylogenetic tree construction using the neighbor-joining algorithm. Phylogenetic trees were visualized using iTOL (82, 83).

## Statistical analysis

To assess statistically significant differences, a two-sample t-test with Welch's correction was used for pairwise comparisons. Statistical analysis was conducted using the ttest_ind() function from the scipy.stats Python module, with the parameter equal_var = False to implement Welch's correction. Significance was evaluated using a threshold of α = 0.05. Differences were considered statistically significant when $P < 0.05$; otherwise, the null hypothesis was not rejected.

## ACKNOWLEDGMENTS

Research reported in this study was supported by the following grants funded to P.A.: Grant Funded by the European Union—Next Generation EU PRIN 2022 PNRR (Project no. P2022LPT3R); Grant Funded by Italian Ministry of University and Research (MUR)—PRIN 2020 (Project no. 202089LLEH); Grant funded by Consorzio Interuniversitario Biotecnologie (DM 587, 08/08/2018; CIB N. 86/19). The funders had no role in study design, data collection and interpretation, or the decision to submit the work for publication. Dr. Marco Costa from the Department of Cultural Heritage, University of Salento, Lecce, Italy, is kindly acknowledged for his support in the GC/MS analysis for the intracellular pyruvate determination.

All authors met ICMJE criteria for authorship and contributed to the study conception and design. Material preparation, experiments, and data collection were performed by A.T., S.C.R., S.M.T., and G.E.D.B. Data analysis was performed by A.T., S.C.R., M.C., G.E.D.B., C.B., and A.P. Project administration and supervision were carried out by P.A. The first draft of the manuscript was written by P.A., and all authors commented on previous versions of the manuscript. All authors read and approved the final manuscript.

## AUTHOR AFFILIATIONS

[1]Department of Biological and Environmental Sciences and Technologies, University of Salento, Lecce, Italy
[2]Department of Experimental Medicine, University of Salento, Lecce, Italy
[3]Laboratory of Analytical and Isotopic Mass Spectrometry, Department of Cultural Heritage, University of Salento, Lecce, Italy

## AUTHOR ORCIDs

Adelfia Talà http://orcid.org/0000-0003-3275-8660
Matteo Calcagnile http://orcid.org/0000-0001-9745-4583
Silvia Caterina Resta http://orcid.org/0000-0001-6102-4151
Giuseppe Egidio De Benedetto http://orcid.org/0000-0002-0832-5470
Cecilia Bucci http://orcid.org/0000-0002-6232-6183
Pietro Alifano http://orcid.org/0000-0003-3768-7275

## FUNDING

| Funder | Grant(s) | Author(s) |
| --- | --- | --- |
| Next Generation EU PRIN 2022 PNRR | P2022LPT3R | Pietro Alifano |
| Italian Ministry of University and Research PRIN 2020 | 202089LLEH | Pietro Alifano |
| Consorzio Interuniversitario Biotecnologie | | Pietro Alifano |

## ADDITIONAL FILES

The following material is available online.

### Supplemental Material

**File S1 (Spectrum00783-25-s0001.xlsx).** Primary data of growth curves of *N. meningitidis* B1940 and isogenic *prpB-* and *prpC*-defective mutants in MCDA-glucose, MCDA-pyruvate, or MCDA-lactate in the absence or presence of different concentrations of propionate.

**File S2 (Spectrum00783-25-s0002.xlsx).** Primary data of growth curves of *N. meningitidis* B1940, B1940ΩprpB, B1940ΩprpC, and complemented strains in MCDA-glucose or MCDA-pyruvate in the absence or presence of 1 mM propionate.

**File S3 (Spectrum00783-25-s0003.xlsx).** Primary data of growth curves of *N. meningitidis* B1940 and isogenic *ackA1-* and *ackA2*-defective mutants in MCDA-glucose in the absence or presence of different concentrations of propionate.

**File S4 (Spectrum00783-25-s0004.xlsx).** Primary data of growth curves of *N. meningitidis* B1940 and isogenic *lstP*-defective mutants in MCDA-glucose, MCDA-pyruvate, or MCDA-lactate in the absence or presence of different concentrations of propionate.

**File S5 (Spectrum00783-25-s0005.xlsx).** Primary data of growth curves of *N. meningitidis* MC58 and isogenic *kbuT*-defective mutants in MCDA-glucose or MCDA-lactate in the absence or presence of different concentrations of propionate.

**File S6 (Spectrum00783-25-s0006.xlsx).** Primary data analysis of *N. meningitidis* strains with BIOLOG Phenotype MicroArray system.

**Supplemental Material (Spectrum00783-25-s0007.pdf).** Fig. S1 to S10 and legends for Files S1 to S6.

### Open Peer Review

**PEER REVIEW HISTORY (review-history.pdf).** An accounting of the reviewer comments and feedback.

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
