## [Reviewer comments · Microbiology Spectrum]

Microbiology Spectrum

Propionic Acid Toxicity and Utilization of α -Ketobutyric Acid in *Neisseria meningitidis* via the Methylcitrate Cycle Under Specific Conditions

Adelfia Talà, Matteo Calcagnile, Silvia Resta, Maurizio Tredici, Giuseppe De Benedetto, Cecilia Bucci, and Pietro Alifano

Corresponding Author(s): Pietro Alifano, Universita del Salento

Review Timeline:

Submission Date:	March 15, 2025
Editorial Decision:	May 31, 2025
Revision Received:	August 5, 2025
Accepted:	September 9, 2025

Editor: Ana-Maria Dragoi

Reviewer(s): The reviewers have opted to remain anonymous.

Transaction Report:

DOI: <https://doi.org/10.1128/spectrum.00783-25>

Re: Spectrum00783-25 (**Propionic Acid Toxicity and Utilization of α -Ketobutyric Acid in *Neisseria meningitidis* via the Methylcitrate Cycle Under Specific Conditions**)

Dear Prof. Pietro Alifano:

Thank you for the privilege of reviewing your work. Below you will find my comments, instructions from the Spectrum editorial office, and the reviewer comments.

The reviewers are generally enthusiastic about the study but suggest major revisions, including the inclusion of key controls and reformatting the figures to make them clearer. I recommend that, should you choose to resubmit, you break down the figures and provide more detail.

Revision Guidelines

Sincerely,
Ana-Maria Dragoi
Editor
Microbiology Spectrum

Reviewer #1 (Comments for the Author):

In this paper Tala et al. report new information about propionate toxicity and the novel discovery of the function of a gene product involved in the utilization of α -ketobutyrate in the human pathogen *Neisseria meningitidis*. The conclusions are well supported by

the data, but the presentation suffers from data overcrowding, which makes the reading of figures very difficult because the font is extremely small even when enlarged on a computer. Below I respectfully offer suggestions for improving the manuscript. I enjoyed reading your work, but it was an unnecessarily difficult read.

1. Importance. Lines 47-50. I suggest the following rewording: In this study, we revealed an unexpected difference in the sensitivity of meningococci to propionate when grown with different carbon sources. We also characterized the function of a gene located within the *prp* operon that encoded a transporter of α -ketobutyrate, an α -keto acid abundant in host cells. Note: at the pH of the medium you have the salt, not the acid form of this compound, hence the use of ' α -ketobutyrate' in lieu of ' α -ketobutyric acid'.
2. Introduction. This section of the manuscript is too long and must be abbreviated. There is too much detailed information that can be eliminated using citations.
3. Line 90. Substitute 'via' for 'from'
4. Line 145. Delete 'the' before growth
5. Line 150. Substitute 'did not grow' for 'were unable to grow'.
6. Line 152. Write 'suggested', not 'suggests'. Use the past tense throughout the manuscript.
7. Line 158. Substitute 'grew' for 'were able to grow'.
8. Line 171. Delete 'were able to' and just write '*prpB* and *prpC* also grew'
9. Line 173. Delete 'were able to' and change the tense from grow to grew.
10. Line 186. Typo: 1,6-biphosphatase not 1,6-bifosphatase.
11. Line 203-4. The authors cannot say that propionate toxicity was due to Pr-CoA and its metabolism, but they can say it may be due to Pr-CoA or its metabolism.
12. Line 205. Please state earlier that Nm has two bona fide acetate kinases. '*ackA2* and '*ackA1*' are incorrect genetic nomenclature. Numbers refer to alleles of the gene. When there are more than one gene function, in this case *ack*, the names should be *ackA*, *ackB*, not *ackA1* and *ackA2*. I am sure there is nothing that can be done here but in the future sticking to genetic convention would be welcome.
13. Line 225-227. Here the authors assume that the *prpB* mutant strain accumulates 2MC while the *prpC* mutant does not. That is not shown in this paper, but if this information is known, it should be cited. The absence of PrpC would accumulate PrCoA that could be used by citrate synthase resulting in a toxic 2-MC isomer. I think the authors should soften the wording here to avoid biasing the reader. Alternatively, if the information about 2-MC accumulation in the *prpB* strain has been reported, please cite that work.
14. Line 302. Substitute 'low cell density' for 'the early growing'.
15. Line 302-303. After the parenthesis, I suggest you write: 'cultures of the wild-type MC58 and *kbuT*-defective strains'
16. Discussion. I suggest the use of Fig 6C in this section to help the reader follow the arguments. Make it big and easy to read.
17. Line 355-356. Modify these lines. I suggest: 'the role of MCC pathway is to assimilate rather than detoxify propionate.'
18. Line 378. The authors did not quantify Pr-CoA levels therefore this statement is not substantiated by any evidence. Please re-phrase.
19. Line 382. Gluconeogenic, not gluconeogenetic.
20. Lines 388-389. You state: 'the pyruvate dehydrogenase complex is allosterically stimulated by pyruvate'. Is this true for the NmPDH complex? If this statement is an extrapolation from work in a different organism, please alert the reader by softening the wording.
21. Line 421. Propionate, not propanoate. I know it is the same thing, but for consistency reasons, please change it to propionate.
22. Figures. Please make the fonts larger. If you must break the figures that is fine but as they are, the fonts are too small even after zooming in with a computer. Fig. 8 is just impossible to read. What is the point of writing names of microorganisms that are unreadable. I know you are referring to different strains of *Neisseria* but it would serve a better purpose to break up the figures and enlarge the fonts.
23. Figure 5. Results in figure A can be readily incorporated into the text; the tables in figure 6B are too small to read; figure 6C should be enlarged and used in the Discussion section; the growth curves in figure 6D should be summarized in a table adjacent to bar figure.
24. Figure S6 is has excruciatingly small font

Reviewer #2 (Comments for the Author):

This manuscript, titled "Propionic Acid Toxicity and Utilization of α -Ketobutyric Acid in *Neisseria meningitidis* via the Methylcitrate Cycle Under Specific Conditions" investigates propionate toxicity and use of the methylcitrate cycle (MCC), a pathway unique to *Neisseria Meningitidis* (NM) among the pathogenic *Neisseriaceae*. NM is only able to grow on select carbon sources - glucose, pyruvate, and lactate. The authors observed that the toxicity of propionic acid is dependent on the carbon source NM is grown on. Propionic acid is toxic to Nm grown in the medium MCDA containing either glucose or pyruvate, but not lactate. This toxicity is somewhat dependent on the activity of the MCC. In MCDA-glucose or MCDA-pyruvate, Nm mutants unable to use the MCC are more sensitive to low concentrations of propionic acid. High concentrations of propionic acid are toxic regardless of MCC activity under these conditions.

The authors observed that this toxicity is also media dependent - as propionic acid is not toxic at high concentrations to Nm grown in CDM-glucose or CDM-pyruvate. The authors determined that this alleviation of toxicity is due to the presence of NaHCO₃ in CDM. Addition of NaHCO₃ to MCDA -glucose alleviated propionate toxicity. However, addition of NaHCO₃ to MCDA -pyruvate only alleviated propionate toxicity for WT Nm. MCC cycle mutants were still sensitive to propionate in MCDA-

pyruvate.

The authors describe multiple possible mechanisms for propionate toxicity, including: decreased intracellular pH, anion accumulation, dissipation of proton motive force, and accumulation of propionyl-CoA. Propionyl-CoA toxicity can be due to inhibition of PDH, inhibition of citrate synthase, conversion into the toxic biproduct 2-methyl-citrate, depletion of the key metabolic intermediate oxaloacetate, homocysteine accumulation, or inactivation of the alanine racemase.

To differentiate these many potential mechanisms, the authors generated a mutant deficient in propionyl coA synthesis - ackA2. The authors found that an ackA2 mutant was resistant to propionate when grown in MCDA-pyruvate, suggesting that synthesis of propionyl-CoA drives toxicity in this condition.

Importantly however, the authors do not observe a rescue in toxicity with the ackA2 mutant in MCDA-glucose. The authors postulate that this is due to inactivity of ackA2 during growth on glucose, and use of ackA1 instead, though the authors do not test an ackA1 mutant to confirm this.

The authors demonstrate that glutathione levels are not altered in MCC mutants compared to WT, which they claim is suggestive that thiol metabolism is not disrupted in these mutants.

The authors show that propionate toxicity is not affected in a lactate permease mutant, suggesting that LctP does not transport propionate, as sometimes occurs in other organisms.

Lastly, the authors demonstrate that in NM, there is a variable gene cluster, encoding different accessory proteins that may be related to the MCC cycle. The authors characterize one such protein, a predicted transporter which they name KbuT. Using a biolog array, followed by individual growth assays, the authors demonstrate that a kbuT mutant is more sensitive to a-ketobutyrate, which they speculate is due to an inability to transport a-ketobutyrate. They further demonstrate that a-ketobutyrate is not consumed from the growth media in a kbuT mutant.

While the observations are interesting, the studies, as presented, are quite confusing to read and not presented in a systematic fashion, particularly in the first section "the toxicity of propionate depends on the main carbon source for meningococcal growth". Furthermore, the studies are lacking in mechanistic detail which would help clarify the biology being presented. The manuscript could be greatly improved by a substantial rewrite of the document and the inclusion of several key controls that are currently missing.

Major Comments:

1. As the authors know, *Neisseria* species are remarkably genetically malleable due in part to frequent phase and antigenic variability. Failure to complement genetic mutations has led to incorrect conclusions in the past (PMID: 23977246.) All of the phenotypes demonstrated in the manuscript have not been genetically complemented, which is essential in this species.
2. The authors describe multiple possible mechanisms for propionate toxicity, however only some of these are conclusively ruled out. For example: the authors speculate that ackA1 and 2 drives accumulation of propionyl-CoA but only mutations in ackA2 were tested.
3. Downstream of propionic acid accumulation, the authors then attempt to rule out some potential mechanisms of toxicity. However, the support for these is fairly weak. For example: The authors speculate that toxicity of propionate is not due to depletion of oxaloacetate, due to the similarity in phenotype between prpC, which directly consumes oxaloacetate to generate 2-methylcitrate from propionyl-coA, and prpB, which works further downstream in the methylcitrate cycle. However as these reactions work in a sequential pathway, this is not strong evidence.
4. The authors later speculate that inhibition of pyruvate dehydrogenase may be the mechanism of toxicity. The authors further speculate that growth on glucose and pyruvate may result in lower accumulation of intracellular pyruvate, compared to lactate, leading to enhanced inhibition of PDH. Some potential suggestions to help support this model include: Evidence could be obtained to support this proposed mechanism by measuring intracellular pyruvate levels, for which colorimetric kits are available. Additionally, if propionate is inhibiting PDH, and function of the TCA cycle to induce toxicity - one would predict that the NAD/NADH ratio of the cell would be altered in the presence of propionate when Nm is grown on glucose or pyruvate. However the NAD/NADH ratio would be unaffected when Nm is grown on lactate, which can be used as an alternative electron donor in *Neisseria* through LdhD and LldD.
5. The authors use OD as a proxy for bacterial burden and examine toxicity. CFU counts would be helpful here - does propionic acid actively kill Nm or just prevent growth?

Minor comments:

1. The manuscript has multiple typographical and grammatical errors, although these don't drastically affect understanding of the content.
2. Many of the figures are small and hard to read. Figure 6B is nearly illegible. I would recommend increasing the font sizes to at least 8, but ideally no smaller than 10 point.
3. Statistics are applied sparingly, and only used in figure 2B. No information on the statistical tests used is provided.
4. The schematic provided in Figure 6C would be very helpful to readers if it was provided early in the manuscript, and should also be much larger.

Review of Ms: Spectrum00783-25

In this paper Tala *et al.* report new information about propionate toxicity and the novel discovery of the function of a gene product involved in the utilization of α -ketobutyrate in the human pathogen *Neisseria meningitidis*. The conclusions are well supported by the data, but the presentation suffers from data overcrowding, which makes the reading of figures very difficult because the font is extremely small even when enlarged on a computer. Below I respectfully offer suggestions for improving the manuscript. I enjoyed reading your work, but it was an unnecessarily difficult read.

1. Importance. Lines 47-50. I suggest the following rewording: In this study, we revealed an unexpected difference in the sensitivity of meningococci to propionate when grown with different carbon sources. We also characterized the function of a gene located within the *prp* operon that encoded a transporter of α -ketobutyrate, an α -keto acid abundant in host cells. Note: at the pH of the medium you have the salt, not the acid form of this compound, hence the use of ' α -ketobutyrate' in lieu of ' α -ketobutyric acid'.
2. Introduction. This section of the manuscript is too long and must be abbreviated. There is too much detailed information that can be eliminated using citations.
3. Line 90. Substitute 'via' for 'from'
4. Line 145. Delete 'the' before growth
5. Line 150. Substitute 'did not grow' for 'were unable to grow'.
6. Line 152. Write 'suggested', not 'suggests'. Use the past tense throughout the manuscript.
7. Line 158. Substitute 'grew' for 'were able to grow'.
8. Line 171. Delete 'were able to' and just write '*prpB* and *prpC* also grew'
9. Line 173. Delete 'were able to' and change the tense from grow to grew.
10. Line 186. Typo: 1,6-biphosphatase not 1,6-bifosphatase.
11. Line 203-4. The authors cannot say that propionate toxicity was due to Pr-CoA and its metabolism, but they can say it may be due to Pr-CoA or its metabolism.
12. Line 205. Please state earlier that *Nm* has two *bona fide* acetate kinases. '*ackA2* and '*ackA1*' are incorrect genetic nomenclature. Numbers refer to alleles of the gene. When there are more than one gene function, in this case *ack*, the names should be *ackA*, *ackB*, *not ackA1* and *ackA2*. I am sure there is nothing that can be done here but in the future sticking to genetic convention would be welcome.
13. Line 225-227. Here the authors assume that the *prpB* mutant strain accumulates 2-MC while the *proc* mutant does not. That is not shown in this paper, but if this information is known, it should be cited. The absence of PrpC would accumulate Pr-CoA that could be used by citrate synthase resulting in a toxic 2-MC isomer. I think the authors should soften the wording here to avoid biasing the reader. Alternatively, if the information about 2-MC accumulation in the *prpB* strain has been reported, please cite that work.
14. Line 302. Substitute 'low cell density' for 'the early growing'.

15. Line 302-303. After the parenthesis, I suggest you write: 'cultures of the wild-type MC58 and *kbuT*-defective strains'
16. Discussion. I suggest the use of Fig 6C in this section to help the reader follow the arguments. Make it big and easy to read.
17. Line 355-356. Modify these lines. I suggest: 'the role of MCC pathway is to assimilate rather than detoxify propionate.'
18. Line 378. The authors did not quantify Pr-CoA levels therefore this statement is not substantiated by any evidence. Please re-phrase.
19. Line 382. Gluconeogenic, not gluconeogenetic.
20. Lines 388-389. You state: 'the pyruvate dehydrogenase complex is allosterically stimulated by pyruvate'. Is this true for the *Nm*PDH complex? If this statement is an extrapolation from work in a different organism, please alert the reader by softening the wording.
21. Line 421. Propionate, not propanoate. I know it is the same thing, but for consistency reasons, please change it to propionate.
22. Figures. Please make the fonts larger. If you must break the figures that is fine but as they are, the fonts are too small even after zooming in with a computer. Fig. 8 is just impossible to read. What is the point of writing names of microorganisms that are unreadable. I know you are referring to different strains of *Neisseria* but it would serve a better purpose to break up the figures and enlarge the fonts.
23. Figure 5. Results in figure A can be readily incorporated into the text; the tables in figure 6B are too small to read; figure 6C should be enlarged and used in the Discussion section; the growth curves in figure 6D should be summarized in a table adjacent to bar figure.
24. Figure S6 is has excruciatingly small font

This manuscript, titled “Propionic Acid Toxicity and Utilization of α -Ketobutyric Acid in *Neisseria meningitidis* via the Methylcitrate Cycle Under Specific Conditions” investigates propionate toxicity and use of the methylcitrate cycle (MCC), a pathway unique to *Neisseria Meningitidis* (NM) among the pathogenic Neisseriaceae. NM is only able to grow on select carbon sources – glucose, pyruvate, and lactate. The authors observed that the toxicity of propionic acid is dependent on the carbon source NM is grown on. Propionic acid is toxic to Nm grown in the medium MCDA containing either glucose or pyruvate, but not lactate.

This toxicity is somewhat dependent on the activity of the MCC. In MCDA-glucose or MCDA-pyruvate, Nm mutants unable to use the MCC are more sensitive to low concentrations of propionic acid. High concentrations of propionic acid are toxic regardless of MCC activity under these conditions.

The authors observed that this toxicity is also media dependent – as propionic acid is not toxic at high concentrations to Nm grown in CDM-glucose or CDM-pyruvate. The authors determined that this alleviation of toxicity is due to the presence of NaHCO₃ in CDM. Addition of NaHCO₃ to MCDA -glucose alleviated propionate toxicity. However, addition of NaHCO₃ to MCDA -pyruvate only alleviated propionate toxicity for WT Nm. MCC cycle mutants were still sensitive to propionate in MCDA-pyruvate.

The authors describe multiple possible mechanisms for propionate toxicity, including: decreased intracellular pH, anion accumulation, dissipation of proton motive force, and accumulation of propionyl-CoA. Propionyl-CoA toxicity can be due to inhibition of PDH, inhibition of citrate synthase, conversion into the toxic biproduct 2-methyl-citrate, depletion of the key metabolic intermediate oxaloacetate, homocysteine accumulation, or inactivation of the alanine racemase.

To differentiate these many potential mechanisms, the authors generated a mutant deficient in propionyl coA synthesis – ackA2. The authors found that an ackA2 mutant was resistant to propionate when grown in MCDA-pyruvate, suggesting that synthesis of propionyl-CoA drives toxicity in this condition.

Importantly however, the authors do not observe a rescue in toxicity with the ackA2 mutant in MCDA-glucose. The authors postulate that this is due to inactivity of ackA2 during growth on glucose, and use of ackA1 instead, though the authors do not test an ackA1 mutant to confirm this.

The authors demonstrate that glutathione levels are not altered in MCC mutants compared to WT, which they claim is suggestive that thiol metabolism is not disrupted in these mutants.

The authors show that propionate toxicity is not affected in a lactate permease mutant, suggesting that LctP does not transport propionate, as sometimes occurs in other organisms.

Lastly, the authors demonstrate that in NM, there is a variable gene cluster, encoding different accessory proteins that may be related to the MCC cycle. The authors characterize one such protein, a predicted transporter which they name KbuT. Using a biollog array, followed by individual growth assays, the authors demonstrate that a kbuT mutant is more sensitive to α -ketobutyrate, which they speculate is due to an inability to transport α -ketobutyrate. They further demonstrate that α -ketobutyrate is not consumed from the growth media in a kbuT mutant.

While the observations are interesting, the studies, as presented, are quite confusing to read and not presented in a systematic fashion, particularly in the first section “the toxicity of propionate depends on the main carbon source for meningococcal growth”. Furthermore, the studies are lacking in mechanistic detail which would help clarify the biology being presented. The manuscript could be greatly improved by a substantial rewrite of the document and the inclusion of several key controls that are currently missing. I am much more enthusiastic about the second half of the manuscript, detailing the newly discovered function of KbuT as an α -ketobutyrate transporter and its distribution in *Neisseria* species, which is better supported.

Major Comments:

1. As the authors know, *Neisseria* species are remarkably genetically malleable due in part to frequent phase and antigenic variability. Failure to complement genetic mutations has led to incorrect conclusions in the past (PMID: 23977246.) All of the phenotypes demonstrated in the manuscript have not been genetically complemented, which is essential in this species.
2. The authors describe multiple possible mechanisms for propionate toxicity, however only some of these are conclusively ruled out. For example: the authors speculate that *ackA1* and *ackA2* drives accumulation of propionyl-CoA but only mutations in *ackA2* were tested.
3. Downstream of propionic acid accumulation, the authors then attempt to rule out some potential mechanisms of toxicity. However, the support for these is fairly weak. For example: The authors speculate that toxicity of propionate is not due to depletion of oxaloacetate, due to the similarity in phenotype between *prpC*, which directly consumes oxaloacetate to generate 2-methylcitrate from propionyl-coA, and *prpB*, which works further downstream in the methylcitrate cycle. However as these reactions work in a sequential pathway, this is not strong evidence.
4. The authors later speculate that inhibition of pyruvate dehydrogenase may be the mechanism of toxicity. The authors further speculate that growth on glucose and pyruvate may result in lower accumulation of intracellular pyruvate, compared to lactate, leading to enhanced inhibition of PDH. Some potential suggestions to help support this model include: Evidence could be obtained to support this proposed mechanism by measuring intracellular pyruvate levels, for which colorimetric kits are available. Additionally, if propionate is inhibiting PDH, and function of the TCA cycle to induce toxicity – one would predict that the NAD/NADH ratio of the cell would be altered in the presence of propionate when *Nm* is grown on glucose or pyruvate. However the NAD/NADH ratio would be unaffected when *Nm* is grown on lactate, which can be used as an alternative electron donor in *Neisseria* through *LdhD* and *LldD*.
5. The authors use OD as a proxy for bacterial burden and examine toxicity. CFU counts would be helpful here – does propionic acid actively kill *Nm* or just prevent growth?

Minor comments:

1. The manuscript has multiple typographical and grammatical errors, although these don't drastically affect understanding of the content.
2. Many of the figures are small and hard to read. Figure 6B is nearly illegible. I would recommend increasing the font sizes to at least 8, but ideally no smaller than 10 point.
3. Statistics are applied sparingly, and only used in figure 2B. No information on the statistical tests used is provided.

Responses to Reviewers of Ms: Spectrum00783-25

REVIEWER 1

In this paper Tala *et al.* report new information about propionate toxicity and the novel discovery of the function of a gene product involved in the utilization of a-ketobutyrate in the human pathogen *Neisseria meningitidis*. The conclusions are well supported by the data, but the presentation suffers from data overcrowding, which makes the reading of figures very difficult because the font is extremely small even when enlarged on a computer. Below I respectfully offer suggestions for improving the manuscript. I enjoyed reading your work, but it was an unnecessarily difficult read.

1. Importance. Lines 47-50. I suggest the following rewording: In this study, we revealed an unexpected difference in the sensitivity of meningococci to propionate when grown with different carbon sources. We also characterized the function of a gene located within the *prp* operon that encoded a transporter of a-ketobutyrate, an a-keto acid abundant in host cells. Note: at the pH of the medium you have the salt, not the acid form of this compound, hence the use of 'a-ketobutyrate' in lieu of 'a-ketobutyric acid'.

R. We thank the reviewer for this suggestion. The text has been changed as suggested (pg. 3, ln. 46-49). Where applicable, "α-ketobutyrate" has been used in place of "α-ketobutyric acid" in the manuscript.

2. Introduction. This section of the manuscript is too long and must be abbreviated. There is too much detailed information that can be eliminated using citations.

R. The Introduction has been shortened. We have eliminated or shortened non-essential sentences while leaving citations. (To avoid compromising the readability of the text, the deleted sentences have not been indicated, but the points where the deletions occurred have been highlighted in red).

3. Line 90. Substitute 'via' for 'from'

R. Corrected (pg. 4, ln. 77).

4. Line 145. Delete 'the' before growth

R. Corrected (pg.7, ln. 128).

5. Line 150. Substitute 'did not grow' for 'were unable to grow'.

R. Corrected (pg.7, ln. 133).

6. Line 152. Write 'suggested', not 'suggests'. Use the past tense throughout the manuscript.

R. Corrected (pg.7, ln. 136).

7. Line 158. Substitute 'grew' for 'were able to grow'.

R. Corrected (pg.8, ln. 155).

8. Line 171. Delete 'were able to' and just write '*prpB* and *prpC* also grew'

R. Corrected (pg.8, ln. 146).

9. Line 173. Delete 'were able to' and change the tense from grow to grew.

R. Corrected (pg.8, ln. 160).

10. Line 186. Typo: 1,6-biphosphatase not 1,6-bifosphatase.
R. Corrected (pg.9, ln. 171).

11. Line 203-4. The authors cannot say that propionate toxicity was due to Pr-CoA and its metabolism, but they can say it may be due to Pr-CoA or its metabolism.
R. Corrected (pg.10, ln. 187).

12. Line 205. Please state earlier that *Nm* has two *bona fide* acetate kinases. ‘*ackA2* and ‘*ackA1*’ are incorrect genetic nomenclature. Numbers refer to alleles of the gene. When there are more than one gene function, in this case *ack*, the names should be *ackA*, *ackB*, not *ackA1* and *ackA2*. I am sure there is nothing that can be done here but in the future sticking to genetic convention would be welcome.
R. The manuscript text has been modified according to the reviewer's helpful suggestion (pg. 10, ln. 190-191). Regarding the nomenclature of the ack genes, we agree with the reviewer, however we preferred not to introduce further changes in this manuscript considering the already confusing nomenclature previously assigned.

13. Line 225-227. Here the authors assume that the *prpB* mutant strain accumulates 2-MC while the *prpC* mutant does not. That is not shown in this paper, but if this information is known, it should be cited. The absence of PrpC would accumulate Pr-CoA that could be used by citrate synthase resulting in a toxic 2-MC isomer. I think the authors should soften the wording here to avoid biasing the reader. Alternatively, if the information about 2-MC accumulation in the *prpB* strain has been reported, please cite that work.
*R. We thank the reviewer for this comment. The text has been edited and information on the accumulation of 2-MC (and 2-MIC) in a *prpB* mutant of *P. aeruginosa* has been reported. We have toned down the wording to avoid bias as suggested and added a sentence on the possibility that propionate toxicity may be mediated by a toxic 2-MC isomer, along with new findings on intracellular pyruvate levels (pg. 10, ln. 212-221).*

14. Line 302. Substitute ‘low cell density’ for ‘the early growing’.
R. Corrected (pg.14, ln. 303).

15. Line 302-303. After the parenthesis, I suggest you write: ‘cultures of the wild-type MC58 and *kbuT*-defective strains’
R. Corrected (pg.14, ln. 303-304).

16. Discussion. I suggest the use of Fig 6C in this section to help the reader follow the arguments. Make it big and easy to read.
R. Figure 6C (now Figure 4) has been enlarged and used more extensively in discussion.

17. Line 355-356. Modify these lines. I suggest: ‘the role of MCC pathway is to assimilate rather than detoxify propionate.’
R. Corrected as suggested (pg.16, ln. 355).

18. Line 378. The authors did not quantify Pr-CoA levels therefore this statement is not substantiated by any evidence. Please re-phrase.
R. The text has been reworded (pg. 17, ln. 376-377).

19. Line 382. Gluconeogenic, not gluconeogenetic.
R. Corrected (pg. 17, ln. 381).

20. Lines 388-389. You state: 'the pyruvate dehydrogenase complex is allosterically stimulated by pyruvate'. Is this true for the *Nm*PDH complex? If this statement is an extrapolation from work in a different organism, please alert the reader by softening the wording.

R. We thank the reviewer for this comment. Allosteric regulation of PHD complex by pyruvate was demonstrate in several proteobacteria, such as E. coli and A. vinelandii. As suggested, we have alerted the reader about the extrapolation in N. meningitidis (pg. 17, ln. 388-390).

21. Line 421. Propionate, not propanoate. I know it is the same thing, but for consistency reasons, please change it to propionate.

R. Corrected (pg. 19, ln. 428).

22. Figures. Please make the fonts larger. If you must break the figures that is fine but as they are, the fonts are too small even after zooming in with a computer. Fig. 8 is just impossible to read. What is the point of writing names of microorganisms that are unreadable. I know you are referring to different strains of *Neisseria* but it would serve a better purpose to break up the figures and enlarge the fonts.

R. All figures have been rearranged with larger fonts. In some cases, the illustrated material has been divided into multiple figures and/or supplementary figures. Figure 8 (now Fig. 10) has been simplified by eliminating panel A and enlarging panel B, enlarging the fonts where possible

23. Figure 5. Results in figure A can be readily incorporated into the text; the tables in figure 6B are too small to read; figure 6C should be enlarged and used in the Discussion section; the growth curves in figure 6D should be summarized in a table adjacent to bar figure.

R. Figure 6 (now Figure 8) has been rearranged. Panel A has been moved to Supplementary material (Fig. S8C). Figure 6B (now Figure 8A) has been simplified by using only a portion and increasing the fonts, while the overall data has been reported in Supplementary File S6.

However, we left panel D, enlarging the fonts.

REVIEWER 2

This manuscript, titled “Propionic Acid Toxicity and Utilization of α -Ketobutyric Acid in *Neisseria meningitidis* via the Methylcitrate Cycle Under Specific Conditions” investigates propionate toxicity and use of the methylcitrate cycle (MCC), a pathway unique to *Neisseria meningitidis* (NM) among the pathogenic Neisseriaceae. NM is only able to grow on select carbon sources – glucose, pyruvate, and lactate. The authors observed that the toxicity of propionic acid is dependent on the carbon source NM is grown on. Propionic acid is toxic to Nm grown in the medium MCDA containing either glucose or pyruvate, but not lactate.

This toxicity is somewhat dependent on the activity of the MCC. In MCDA-glucose or MCDA-pyruvate, Nm mutants unable to use the MCC are more sensitive to low concentrations of propionic acid. High concentrations of propionic acid are toxic regardless of MCC activity under these conditions.

The authors observed that this toxicity is also media dependent – as propionic acid is not toxic at high concentrations to Nm grown in CDM-glucose or CDM-pyruvate. The authors determined that this alleviation of toxicity is due to the presence of NaHCO₃ in CDM. Addition of NaHCO₃ to MCDA-glucose alleviated propionate toxicity. However, addition of NaHCO₃ to MCDA-pyruvate only alleviated propionate toxicity for WT Nm. MCC cycle mutants were still sensitive to propionate in MCDA-pyruvate.

The authors describe multiple possible mechanisms for propionate toxicity, including: decreased intracellular pH, anion accumulation, dissipation of proton motive force, and accumulation of propionyl-CoA. Propionyl-CoA toxicity can be due to inhibition of PDH, inhibition of citrate synthase, conversion into the toxic biproduct 2-methyl-citrate, depletion of the key metabolic intermediate oxaloacetate, homocysteine accumulation, or inactivation of the alanine racemase.

To differentiate these many potential mechanisms, the authors generated a mutant deficient in propionyl coA synthesis – ackA2. The authors found that an ackA2 mutant was resistant to propionate when grown in MCDA-pyruvate, suggesting that synthesis of propionyl-CoA drives toxicity in this condition.

Importantly however, the authors do not observe a rescue in toxicity with the ackA2 mutant in MCDA-glucose. The authors postulate that this is due to inactivity of ackA2 during growth on glucose, and use of ackA1 instead, though the authors do not test an ackA1 mutant to confirm this.

The authors demonstrate that glutathione levels are not altered in MCC mutants compared to WT, which they claim is suggestive that thiol metabolism is not disrupted in these mutants.

The authors show that propionate toxicity is not affected in a lactate permease mutant, suggesting that LctP does not transport propionate, as sometimes occurs in other organisms.

Lastly, the authors demonstrate that in NM, there is a variable gene cluster, encoding different accessory proteins that may be related to the MCC cycle. The authors characterize one such protein, a predicted transporter which they name KbuT. Using a biolog array, followed by individual growth assays, the authors demonstrate that a kbuT mutant is more sensitive to α -ketobutyrate, which they speculate is due to an inability to transport α -ketobutyrate. They further demonstrate that α -ketobutyrate is not consumed from the growth media in a kbuT mutant.

While the observations are interesting, the studies, as presented, are quite confusing to read and not presented in a systematic fashion, particularly in the first section “the toxicity of propionate depends on the main carbon source for meningococcal growth”. Furthermore, the studies are lacking in mechanistic detail which would help clarify the biology being presented. The manuscript

could be greatly improved by a substantial rewrite of the document and the inclusion of several key controls that are currently missing. I am much more enthusiastic about the second half of the manuscript, detailing the newly discovered function of KbuT as an α -ketobutyrate transporter and its distribution in *Neisseria* species, which is better supported.

R. We have tried to improve the clarity of the first part of the manuscript, especially in the part of interpreting the data in relation to the mechanistic hypotheses based on the available data, pointing out that some hypotheses need further investigation. We have, in addition, introduced some key control experiments, including some complementation tests, tests with new mutants and tests performed with independent mutants of the single genes to avoid misinterpretation of results, and additional biochemical tests, as detailed below. We have, finally, improved the presentation of the figures, many of which were indeed unreadable due to the fonts used.

Major Comments:

1. As the authors know, *Neisseria* species are remarkably genetically malleable due in part to frequent phase and antigenic variability. Failure to complement genetic mutations has led to incorrect conclusions in the past (PMID: 23977246.) All of the phenotypes demonstrated in the manuscript have not been genetically complemented, which is essential in this species.

*R. We agree with the reviewer on this point. To exclude the possibility that some phenotypes observed in the *prpB*, *prpC*, *ackA1*, *ackA2*, and *kbuT* mutants may have been generated by variations in other genes due to the genetic instability of *N. meningitidis*, some growth phenotypes were evaluated using independent mutants (with the exception of *prpC*, for which a single mutant was obtained), as clarified in the revised manuscript (pg. 7, ln. 119-122; Files S1-S6). Additionally, *prpB* and *prpC* mutations were genetically complemented and the data have been reported (p. 7, ln. 122-125 and ln. 128-129; pg. 22, ln. 510-511; pg. 23, ln. 538-550; new Fig. S2 and File S2).*

2. The authors describe multiple possible mechanisms for propionate toxicity, however only some of these are conclusively ruled out. For example: the authors speculate that *ackA1* and 2 drives accumulation of propionyl-CoA but only mutations in *ackA2* were tested.

*R. We thank the Reviewer for this suggestion. To confirm our hypothesis, we generated an *ackA1*-defective mutant and showed that this mutant grew in MCDA-glucose containing 5 mM propionate (pg 10, ln. 204-211; pg. 17, ln. 368-369; pg. 22, ln. 519, 521, 523, 525, 527, 528, 534; new Fig. S5).*

3. Downstream of propionic acid accumulation, the authors then attempt to rule out some potential mechanisms of toxicity. However, the support for these is fairly weak. For example: The authors speculate that toxicity of propionate is not due to depletion of oxaloacetate, due to the similarity in phenotype between *prpC*, which directly consumes oxaloacetate to generate 2-methylcitrate from propionyl-coA, and *prpB*, which works further downstream in the methylcitrate cycle. However as these reactions work in a sequential pathway, this is not strong evidence.

R. We thank the reviewer for this comment and for pointing out the inaccuracy regarding the possible mechanism involving oxaloacetate depletion. The paragraph on possible mechanisms or toxicity of propionate has been revised, taking also into account the comments of Reviewer 1 (pg. 10, ln. 212-221).

4. The authors later speculate that inhibition of pyruvate dehydrogenase may be the mechanism of toxicity. The authors further speculate that growth on glucose and pyruvate may result in lower accumulation of intracellular pyruvate, compared to lactate, leading to enhanced inhibition of PDH. Some potential suggestions to help support this model include: Evidence could be obtained to support this proposed mechanism by measuring intracellular pyruvate levels, for which colorimetric kits are available. Additionally, if propionate is inhibiting PDH, and function of the TCA cycle to induce toxicity – one would predict that the NAD/NADH ratio of the cell would be altered in the presence of propionate when Nm is grown on glucose or pyruvate. However the NAD/NADH ratio would be unaffected when Nm is grown on lactate, which can be used as an alternative electron donor in Neisseria through LdhD and LldD.

R. We thank the Reviewer for this suggestion. We have determined the intracellular pyruvate levels in bacteria growing in the different media by GC/MS and provide evidence that these levels were decreasing when lactate, glucose and pyruvate were used as main carbon sources, respectively (pg. 11, ln. 229-233; pg. 18, ln 401-407; pg. 25, ln. 590-609; Fig. S6C).

5. The authors use OD as a proxy for bacterial burden and examine toxicity. CFU counts would be helpful here – does propionic acid actively kill Nm or just prevent growth?

R. CFU counts were determined, and the data suggest that propionate does not kill the bacteria but just prevent their growth (pg. 8, ln. 139-142; new Fig. S3).

Minor comments:

1. The manuscript has multiple typographical and grammatical errors, although these don't drastically affect understanding of the content.

R. The manuscript has been revised for typographical and grammatical errors, also based on the comments of reviewer 1.

2. Many of the figures are small and hard to read. Figure 6B is nearly illegible. I would recommend increasing the font sizes to at least 8, but ideally no smaller than 10 point.

R. All figures have been rearranged with larger fonts. In some cases, the illustrated material has been divided into multiple figures and/or supplementary figures. Figure 6B (now Figure 8A) has been simplified by using only a portion and increasing the fonts, while the overall data has been reported in Supplementary File S6.

3. Statistics are applied sparingly, and only used in figure 2B. No information on the statistical tests used is provided.

A. Information on the statistical tests used in this study has been provided (p. 27, ln. 644-649) and the processed data have been included in figures and legends (Fig. 1, Fig. 2, Fig. 3, Fig. 5, Fig. 6, Fig. 8, Fig. S2, Fig. S3, Fig. S6), while row data are reported in supplementary files.

Re: Spectrum00783-25R1 (Propionic Acid Toxicity and Utilization of α -Ketobutyric Acid in *Neisseria meningitidis* via the Methylcitrate Cycle Under Specific Conditions)

Dear Prof. Pietro Alifano:

Your manuscript has been accepted, and I am forwarding it to the ASM production staff for publication. Your paper will first be checked to make sure all elements meet the technical requirements. ASM staff will contact you if anything needs to be revised before copyediting and production can begin. Otherwise, you will be notified when your proofs are ready to be viewed.

Although the paper has been accepted, please ensure that the reviewer's comments are address in the final version.

Sincerely,
Ana-Maria Dragoi
Editor
Microbiology Spectrum

Reviewer #2 (Comments for the Author):

This revised manuscript " Propionic Acid Toxicity and Utilization of α -ketobutyric acid in *Neisseria meningitidis* via the Methylcitrate cycle under specific conditions" represents a significant revision over the original manuscript. The manuscript as a whole is much easier to follow and is presented logically. In general, the authors effectively addressed the scientific and text comments requested by both reviewer 1 and 2. My remaining comments are mostly text based in nature.

Comments

1. The introduction is much improved in terms of clarity of presentation. However I wonder if the background on glucose, lactate, and pyruvate utilization in *Neisseria* is necessary. Indeed lines 76-100 seem largely unnecessary to understand the premise of the paper (for example the background on the glucose transporter glcP and the TCA cycle). Because of this, the introduction of propionic acid and the MCC at line 101 seems like an afterthought in comparison. Expanding on some background on the MCC, propionic acid, kbuT, and α -ketobutyric acid here would help establish its relevance to the reader.

However I also need to note that the text is already quite long - rather than introducing new material entirely, I think several portions of the discussion could be moved to the introduction to better balance the text. For example, the discussion of α -ketobutyric acid in lines 437-439, could be simplified some, and moved to the introduction.

2. In reference to calling propionate "toxic" - line 116 and throughout the text. Since propionate only inhibits growth and does not actively kill the bacteria, it may be more apt to refer to "growth inhibition by propionate" rather than "propionate toxicity".

3. Lines 105-109: This sentence is quite complex. I would separate it into 2-3 sentences to confer each idea more simply.

4. Lines 119-120 should pDE prpB and pDEprpC be italicized, since these are gene names?

5. Lines 120-125: I very much appreciate that the authors addressed my previous comment about complementation of their mutants. However, the description of the bacterial strains used here is quite long and repeats content in the methods section. These lines could be removed entirely. The reference to testing complemented strains in lines 130-131 is sufficient.
6. Line 172: typo - should read "bacterial and fungal systems"
7. Line 244: grammatical error - should read "why were the strains not sensitive to propionate when grown on lactate?"
8. Line 275: a comma is needed after "in 75 genomes,"
9. Line 356: grammatical error - should read "the MCC allows attenuation of"
10. Throughout the text and figures, the symbol omega is used to denote mutants rather than a delta. I am assuming this is meant to denote that these are insertional mutants rather than clean deletions, however I am unfamiliar with this notation. Its possible I am just unaware, however conventionally, I think insertional mutants are written as prpB::erm.

This revised manuscript “ Propionic Acid Toxicity and Utilization of a-ketobutyric acid in *Neisseria meningitidis* via the Methylcitrate cycle under specific conditions” represents a significant revision over the original manuscript. The manuscript as a whole is much easier to follow and is presented logically. In general, the authors effectively addressed the scientific and text comments requested by both reviewer 1 and 2. My remaining comments are mostly text based in nature.

Comments

1. The introduction is much improved in terms of clarity of presentation. However I wonder if the background on glucose, lactate, and pyruvate utilization in *Neisseria* is necessary. Indeed lines 76-100 seem largely unnecessary to understand the premise of the paper (for example the background on the glucose transporter *glcP* and the TCA cycle). Because of this, the introduction of propionic acid and the MCC at line 101 seems like an afterthought in comparison. Expanding on some background on the MCC, propionic acid, *kbuT*, and a-ketobutyric acid here would help establish its relevance to the reader.

However I also need to note that the text is already quite long – rather than introducing new material entirely, I think several portions of the discussion could be moved to the introduction to better balance the text. For example, the discussion of a-ketobutyric acid in lines 437-439, could be simplified some, and moved to the introduction.

2. In reference to calling propionate “toxic” – line 116 and throughout the text. Since propionate only inhibits growth and does not actively kill the bacteria, it may be more apt to refer to “growth inhibition by propionate” rather than “propionate toxicity”.
3. Lines 105-109: This sentence is quite complex. I would separate it into 2-3 sentences to confer each idea more simply.
4. Lines 119-120 should *pDE prpB* and *pDEprpC* be italicized, since these are gene names?
5. Lines 120-125: I very much appreciate that the authors addressed my previous comment about complementation of their mutants. However, the description of the bacterial strains used here is quite long and repeats content in the methods section. These lines could be removed entirely. The reference to testing complemented strains in lines 130-131 is sufficient.
6. Line 172: typo – should read “bacterial and fungal systems”
7. Line 244: grammatical error – should read “why were the strains not sensitive to propionate when grown on lactate?”
8. Line 275: a comma is needed after “in 75 genomes,”
9. Line 356: grammatical error – should read “the MCC allows attenuation of”

10. Throughout the text and figures, the symbol omega is used to denote mutants rather than a delta. I am assuming this is meant to denote that these are insertional mutants rather than clean deletions, however I am unfamiliar with this notation. Its possible I am just unaware, however conventionally, I think insertional mutants are written as *prpB::erm*.